# Activation of the essential kinase PDK1 by phosphoinositide-driven *trans*-autophosphorylation

Aleksandra Levina[1,2], Kaelin D. Fleming [3], John E. Burke [3,4] & Thomas A. Leonard [1,2✉]

3-phosphoinositide-dependent kinase 1 (PDK1) is an essential serine/threonine protein kinase, which plays a crucial role in cell growth and proliferation. It is often referred to as a 'master' kinase due to its ability to activate at least 23 downstream protein kinases implicated in various signaling pathways. In this study, we have elucidated the mechanism of phosphoinositide-driven PDK1 auto-activation. We show that PDK1 *trans*-autophosphorylation is mediated by a PIP$_3$-mediated face-to-face dimer. We report regulatory motifs in the kinase-PH interdomain linker that allosterically activate PDK1 autophosphorylation via a linker-swapped dimer mechanism. Finally, we show that PDK1 is autoinhibited by its PH domain and that positive cooperativity of PIP$_3$ binding drives switch-like activation of PDK1. These results imply that the PDK1-mediated activation of effector kinases, including Akt, PKC, Sgk, S6K and RSK, many of whom are not directly regulated by phosphoinositides, is also likely to be dependent on PIP$_3$ or PI(3,4)P$_2$.

[1] Department of Structural and Computational Biology, Max Perutz Labs, Campus Vienna Biocenter 5, 1030 Vienna, Austria. [2] Department of Medical Biochemistry, Medical University of Vienna, 1090 Vienna, Austria. [3] Department of Biochemistry and Microbiology, University of Victoria, Victoria, BC V8W 2Y2, Canada. [4] Department of Biochemistry and Molecular Biology, The University of British Columbia, Vancouver, BC V6T 1Z3, Canada. ✉email: thomas.leonard@meduniwien.ac.at

PDK1 (3-phosphoinositide-dependent kinase 1) is a serine/threonine-protein kinase that plays a central role in regulating cell growth, survival, and proliferation. Knocking out PDK1 in mice leads to early embryonic lethality[1]. PDK1 overexpression has been observed in breast and ovarian tumors[2,3], and its activity is increased in prion-infected neurons[4]. PDK1 belongs to the AGC kinase family (which derives its nomenclature from protein kinases A, G, and C) that consists of 63 members of which 23 have been reported to be PDK1 substrates, including Akt/protein kinase B (PKB), protein kinase C (PKC), p70 S6 kinase (S6K), and ribosomal S6 kinase (RSK)[5]. As such, it is often referred to as a 'master kinase'. The hyperactivation of Akt, downstream of PDK1, is a hallmark of many human cancers, as well as a number of rare overgrowth disorders and immune diseases[6,7].

PDK1 consists of an N-terminal kinase domain and a C-terminal phosphoinositide-binding PH domain. The PH domain binds to phosphatidylinositol-3,4-bisphosphate (PI(3,4)$P_2$) and phosphatidylinositol-3,4,5-trisphosphate (henceforth abbreviated to $PIP_3$) with low nanomolar affinity[8], allowing PDK1 to be translocated to the plasma membrane upon insulin- or growth-factor-stimulated $PIP_3$ production by phosphoinositide 3-kinase (PI3K). Phosphorylation of a conserved serine (S241) in the activation loop leads to PDK1 activation. Mutation of Ser241 to alanine reduces its activity against Akt by 50-fold while rendering it inactive against Sgk[9]. Activation loop phosphorylation of PDK1 has previously been observed to occur in *E. coli*[9] and with purified protein in vitro[10], suggesting that PDK1 can activate itself by *trans*-autophosphorylation. Consistent with these observations, kinase-dead PDK1 is *trans*-phosphorylated on Ser241 by wild-type PDK1 in vitro[11], PDK1 dimerization has been reported in cells[12,13], and PH domain-mediated dimerization has been reconstituted on $PIP_3$-containing lipid bilayers[14]. However, the molecular mechanisms of lipid-mediated activation by dimerization and *trans*-autophosphorylation remain poorly understood.

Whether PDK1 activation is regulated at all remains controversial. Studies performed on overexpressed PDK1 in different cell lines have reported contradictory results. Some have reported no or minor changes in PDK1 localization in response to insulin[8,15] whereas others have reported insulin-dependent membrane translocation[16] and growth-factor-dependent PDK1 activation[13]. High levels of PDK1 activation loop (S241) phosphorylation that are not affected by insulin stimulation are often observed in PDK1 overexpressed cells[11,17,18]. However, S241 phosphorylation of endogenous PDK1 has been shown to be insulin-dependent in a mouse hypothalamic cell line[19]. These contradictions highlight the difficulties of studying signaling events in cells under conditions of ectopic overexpression, cell population response heterogeneity, and the transient nature of the events themselves. Nevertheless, these observations have led many to conclude that PDK1 is constitutively active in cells and that signal propagation is regulated by a variety of mechanisms at the level of its substrates, including co-localization, allosteric activation, or post-translational modification. These mechanisms, however, beg the question of why PDK1 is regulated by activation loop phosphorylation at all. Moreover, the presence of a constitutively active kinase in cells is likely to lead to spurious off-target phosphorylation events, uncoupled from growth factor signaling, that are incompatible with the coordination of cellular events in a tightly regulated manner in both space and time.

These contradictions prompted us to address the question of how PDK1 activity is regulated at a molecular level. We show that PDK1 activation by *trans*-autophosphorylation is dependent on a specific face-to-face dimer mediated by its αG helix. *Trans*-autophosphorylation is further enhanced by two previously uncharacterized motifs in the kinase-PH interdomain linker of PDK1. Finally, we show that PDK1 exists in an autoinhibited conformation, irrespective of its phosphorylation state, which is directly activated by $PIP_3$. We propose a model of lipid-dependent PDK1 activation in which PH domain-mediated autoinhibition is relieved by $PIP_3$ or PI(3,4)$P_2$ binding, leading to strongly cooperative kinase domain dimerization and switch-like *trans*-autophosphorylation at the membrane. Our findings suggest that PDK1 activation and activity, as well as the activation of 23 downstream kinases, themselves PDK1 substrates, are restricted to $PIP_3$ or PI(3,4)$P_2$-containing membranes.

## Results

### A transient but specific dimer controls PDK1 autophosphorylation.

To investigate the mechanism of PDK1 auto-activation, we first sought to establish suitable construct boundaries of the PDK1 kinase domain that would permit a systematic and quantitative analysis of activation loop autophosphorylation. All constructs employed in this study are defined in Supplementary Table 1 and intact mass spectra for all recombinant proteins are provided in Supplementary Figs. 1-2. Chordates encode a region of 50 amino acids immediately upstream of an alternative splice site that is not conserved or missing in older PDK1 orthologs. This region is predicted to be intrinsically disordered and its function has not been investigated. It is not present in isoform 2 of human PDK1. The reported PDK1 kinase domain comprising residues 51–359 (UniProt O15530-2) exhibited autophosphorylation of a second residue with low efficiency in an in vitro kinase assay (Supplementary Fig. 1a). Since residues 51-71 are disordered in crystal structures of the PDK1 kinase domain, we truncated the N-terminus to create a construct comprising residues 73–359 (Fig. 1a) that undergoes stoichiometric activation loop phosphorylation in vitro (Supplementary Fig. 1b-d). Henceforth, we refer to this construct as PDK1$^{SKD}$ (short kinase domain, where 'short' in this context refers to the C-terminal boundary of the construct).

Both the short kinase domain (PDK1$^{SKD}$) and near-full-length PDK1 (PDK1$^{FL}$, residues 73–556) are monomeric in solution as observed by size exclusion chromatography coupled to multi-angle light scattering (SEC-MALS) (Fig. 1b) at a concentration (2 μM) three orders of magnitude higher than that reported in cells[20], consistent with expectations for a transient interaction. We therefore modeled the PDK1 kinase domain dimer using the ROSETTA symmetric modeling tool[21] using the crystal structure of the monomeric kinase domain (PDB 2BIY). Since the conformation of the activation loop in the context of the dimer is unknown, we deleted the corresponding residues 228–245 from the monomer structure for the purposes of modeling. The best scoring model exhibited a face-to-face arrangement of the two protomers (Fig. 1c) in which the surface of interaction is highly conserved and partially hydrophobic in nature (Supplementary Fig. 3a). Strikingly, the dimer model is superimposable with crystal structures of MEK1 and MEK2 dimers (Supplementary Fig. 3b), as well as the B-RAF:MEK1 complex (Supplementary Fig. 3c), in which the αG helix is found in the dimerization interface. Independent modeling of the kinase domain dimer with AlphaFold2, which employs a deep learning neural network algorithm to combine evolutionary information with known physical and stereochemical constraints[22,23], resulted in an essentially identical arrangement of the two protomers, with a root mean square deviation (r.m.s.d.) of 1.87–2.73 Å overall backbone $C_\alpha$ atoms of both chains for ten models generated with and without template matching (Supplementary Fig. 3d).

In the PDK1 dimer, an invariant tyrosine or phenylalanine (human Y288, Supplementary Fig. 4a) is contributed to the

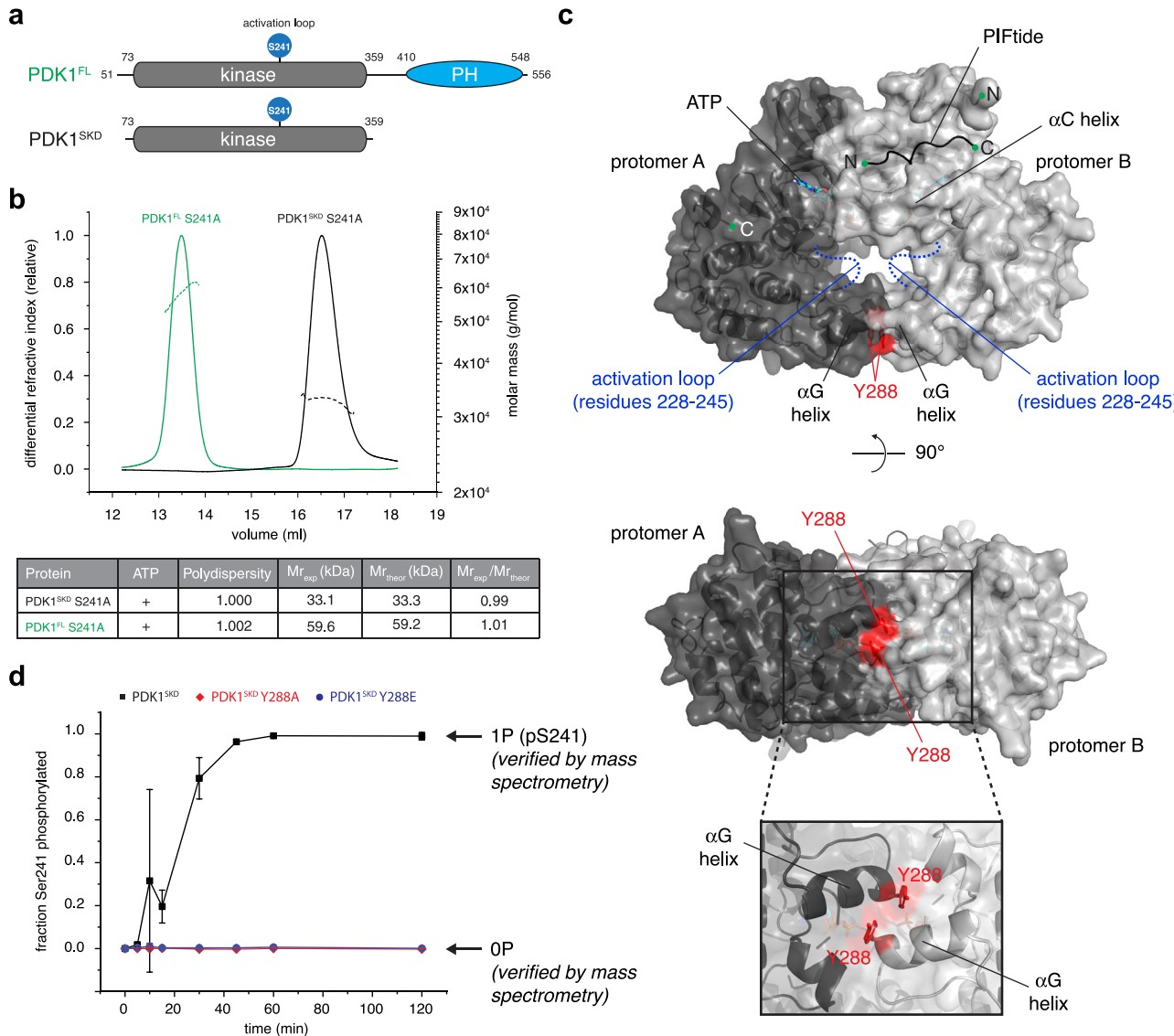

**Fig. 1 A transient, but specific dimer controls PDK1 autophosphorylation. a** Cartoon schematic of PDK1 domain architecture and construct boundaries of PDK1$^{FL}$ (full length) and PDK1$^{SKD}$ (short kinase domain). **b** Size exclusion chromatography coupled to multi-angle light scattering (SEC-MALS) of PDK1$^{FL}$ S241A (green) and PDK1$^{SKD}$ S241A (black). The table summarizes protein polydispersity, theoretical (Mr$_{theor}$) and experimentally determined (Mr$_{exp}$) molecular weights, as well as the calculated oligomeric state (Mr$_{exp}$/Mr$_{theor}$). **c** Model of the PDK1 kinase domain dimer obtained using the ROSETTA symmetric modeling tool. Kinase domains (PDB: 2biy) are shown in dark and light gray. Activation loop residues 228–245 that were removed from the structure for the purpose of modeling are shown in dashed blue lines. The hydrophobic motif of PRK2 (PIFtide) bound to the hydrophobic pocket (PDB: 4rrv) is depicted with a black line. Y288 in the dimer interface is highlighted in red. **d** Radiometric PDK1 autophosphorylation assay. PDK1$^{SKD}$ (closed black squares), PDK1$^{SKD}$ Y288A (closed red diamonds), and PDK1$^{SKD}$ Y288E (closed blue circles). $n = 3$ biologically independent experiments. Data are presented as mean values ± S.D.

interface by the αG helix of each protomer (Fig. 1c). Reported substrate kinases of PDK1 that do not undergo autophosphorylation do not exhibit conservation at this position in their αG helices (Supplementary Fig. 4b). Therefore, to test our in silico model in vitro, we mutated Y288 to either alanine (PDK1$^{SKD}$ Y288A, Supplementary Fig. 1e), thereby removing the side chain, or glutamate (PDK1$^{SKD}$ Y288E, Supplementary Fig. 1f), thereby creating a charge repulsion in the interface. The transient dimerization of wild-type and mutant kinase domains was evaluated with a radiometric autophosphorylation assay using labeled [γ-$^{32}$P] ATP. Mutation of Y288 to either alanine or glutamate resulted in complete abrogation of autophosphorylation under assay conditions in which the wild-type kinase domain is

stoichiometrically phosphorylated on S241 (Fig. 1d, Supplementary Fig. 1d). It is worth noting that expression of both PDK1$^{SKD}$ Y288A and PDK1$^{SKD}$ Y288E with an N-terminal dimeric GST-tag in insect cells resulted in stoichiometrically monophosphorylated proteins (Supplementary Fig. 1g), indicating that mutation of Y288 does not completely abrogate autophosphorylation and can be overcome by artificial dimerization. Moreover, the crystal structure of the kinase domain of PDK1 bearing a Y288G mutation designed to disrupt lattice contacts[24] (Supplementary Fig. 4c) illustrates that mutation of Y288 does not result in any conformational changes in the kinase domain. Together, these results indicate that PDK1 autophosphorylation is dependent on a specific kinase domain dimer centered on Y288.

**PIF pocket occupancy promotes PDK1 *trans*-autophosphorylation**. Typical AGC kinase domains contain a hydrophobic pocket in the N-lobe and a long C-terminal tail containing multiple regulatory motifs, including a hydrophobic motif (HM) with the sequence ϕXXϕ, where ϕ = hydrophobic amino acid and X = any amino acid[25]. Docking of the HM into the hydrophobic pocket promotes the active conformation of the kinase domain and its mutation is associated with reduced kinase activity[26–29]. Some, but not all, AGC kinases also possess a regulatory phosphorylation site immediately C-terminal to the HM. AGC kinases with phosphorylatable HMs or with glutamate substitutions, including Akt, PKC, RSK, S6K, and Sgk, are reported substrates of PDK1[5]. Phosphorylation of the HM of these substrate kinases is reported to drive association with the hydrophobic pocket of PDK1 and consequent activation loop phosphorylation of these substrates[10,30–35].

In contrast to other members of the AGC kinase family, PDK1 does not contain a canonical C-terminal tail. Instead, the kinase domain is followed by a flexible linker and a C-terminal membrane-binding PH domain. Nevertheless, binding of a HM peptide to the hydrophobic pocket of PDK1 leads to conformational changes resulting in higher ATP binding affinity[36] and increased overall stability of the kinase domain[30,37]. The prevailing model of substrate phosphorylation by PDK1 therefore suggests that the phosphorylated HM of the substrate binds into the hydrophobic pocket of PDK1 and allosterically activates PDK1 against the docked substrate. However, the binding of a HM peptide to the hydrophobic pocket has also been shown to promote PDK1 autophosphorylation[10]. We therefore reasoned that PDK1 autoactivation likely depends on a functional HM.

To test whether a functional HM is necessary for kinase domain dimerization and autophosphorylation, we fused PIFtide, a peptide derived from the HM of PRK2 (QEMFRDFDYIADWC) that binds to PDK1 with low nanomolar affinity[38], to the N-terminus of the kinase domain (PDK1[PIF-SKD], Supplementary Fig. 1h). The N-terminus (Q73) is just 8 Å away from the end of the PIFtide sequence (PDB 4RRV), which could be bridged with a simple poly-Gly (G$_5$) linker (Fig. 2a). This construct design ensured constitutive binding of the peptide to the hydrophobic pocket. To avoid autophosphorylation in the presence of ATP, which might disrupt dimerization, we introduced a S241A mutation (Supplementary Fig. 1i). While SEC-MALS revealed PDK1[PIF-SKD] S241A to be monomeric (Fig. 2b, orange curve), even in the presence of ATP (Fig. 2b, light orange curve), a fusion of PIFtide to wild-type PDK1[SKD] resulted in faster autophosphorylation kinetics (Fig. 2c), confirming that occupancy of its hydrophobic pocket promotes PDK1 autophosphorylation.

In order to obtain a stable kinase domain dimer amenable to structural and biophysical analysis, we next fused the PIFtide sequence to the C-terminus of the kinase domain (PDK1[SKD-PIF]) (Fig. 2a). Our in silico model of the kinase domain dimer revealed that the C-terminus of the kinase domain and the first residue of a HM peptide (modeled using PDB 4RRV) docked in the hydrophobic pocket of the opposing protomer were just 29.4 Å apart. We therefore designed a minimal poly Gly-Ser (GS)$_4$ linker that would allow the C-terminal PIFtide to reach into the hydrophobic pocket of the other protomer. We hypothesized that, provided that the linker length is sufficient, the high affinity of the PIFtide towards the hydrophobic pocket should result in its constitutive binding and lead to stable dimer formation. SEC-MALS revealed that PDK1[SKD-PIF] S241A (Supplementary Fig. 1j) is a constitutive dimer in both the absence (Fig. 2b, blue curve) and presence of ATP (Fig. 2b, light blue curve). The presence of ATP resulted in a noticeable reduction in the hydrodynamic radius of PDK1[SKD-PIF] (ΔVe = 0.35 ml, gray arrow, Fig. 2b), consistent with the specific organization of the two activation loops in the center of the dimer and consequent compaction of the particle.

The PDK1[SKD-PIF] dimer was then used to analyze the conformational changes associated with dimer formation by crosslinking coupled to mass spectrometry (XL-MS) using the zero-length, heterobifunctional crosslinker 1-ethyl-3-(3-dimethylaminopropyl)carbodiimide (EDC). We determined dimer- and monomer-specific crosslinks and mapped the crosslinks onto the dimer model (Fig. 2d). In the dimer, we observed 36 novel crosslinks between residues in the C-terminal PIFtide sequence and residues in the N-terminus of PDK1 (Supplementary Table 2) consistent with a face-to-face dimer formed by exchange of the C-terminal PIFtide motifs. We excluded the possibility of these crosslinks being intramolecular on the basis that the linker sequence is too short for the PIFtide motif to bind into the hydrophobic pocket of its own protomer. We also observed striking changes within the activation loop. 51 out of 53 crosslinks in the monomer that were formed between the activation loop and either the N lobe (glycine-rich loop and αB helix) or C lobe (αG and αH helices) as well as a specific crosslink in the catalytic loop of the kinase domain were not present in the dimer (Supplementary Table 3). This result is consistent with the activation loop being highly flexible when not phosphorylated and suggests that it adopts a different conformation in the dimer, which is presumably permissive for *trans*-autophosphorylation.

Further structural details were obtained by hydrogen-deuterium exchange-mass spectrometry (HDX-MS). HDX-MS reports the exchange rate of amide hydrogens and, as the primary determinant of amide exchange is the involvement in secondary structure, it acts as a probe to measure changes in secondary structure dynamics[39]. We compared the dimeric PDK1[SKD-PIF] to monomeric PDK1[SKD] and mapped the differences in deuterium incorporation onto the dimer model (Fig. 2e-f). In the N-lobe, the largest decrease in exchange is observed around the hydrophobic pocket (αC and αB helixes), consistent with stabilization of the N-lobe secondary structure by the PIFtide hydrophobic motif. Neighboring regions, as well as the ATP binding pocket, catalytic loop, and activation loop, are also protected in the dimer, observations that support known conformational changes in AGC protein kinases upon hydrophobic motif pocket occupancy. Distinguishing between conformational changes induced by binding of the PIFtide motif into the hydrophobic pocket and those induced by dimerization is, however, not possible on the basis of this data. Curiously, we did not observe any changes in the αG helix associated with dimerization, which most likely reflects the fact that the αG helix is fully ordered in the monomeric kinase domain (Supplementary Fig. 4c), as well as the kinase domain dimer model (Fig. 1c) and, consequently, secondary structure changes in the αG helix upon dimerization, are unlikely. Finally, we observed increased exchange of the αH-αI loop in PDK1[SKD-PIF] suggesting that specific interactions present in the monomeric kinase domain are lost upon dimerization (Fig. 2e, yellow). Superimposition of the activation loop from the structure of PDK1 bearing a S241 to alanine mutation (Fig. 2g, red), revealed that the tip of the activation loop contacts the deprotected region, an observation that is consistent with our XL-MS data and confirms a conformational change of the activation loop during dimerization, an event which is a prerequisite for *trans*-autophosphorylation.

In conclusion, PDK1 autophosphorylation is promoted by occupancy of its hydrophobic pocket, and kinase domain dimerization leads to remodeling of the activation loop for *trans*-autophosphorylation.

**PDK1 encodes a hydrophobic motif that drives *trans*-autophosphorylation**. Since PDK1 autophosphorylation is promoted by binding of PIFtide to its hydrophobic pocket, we next

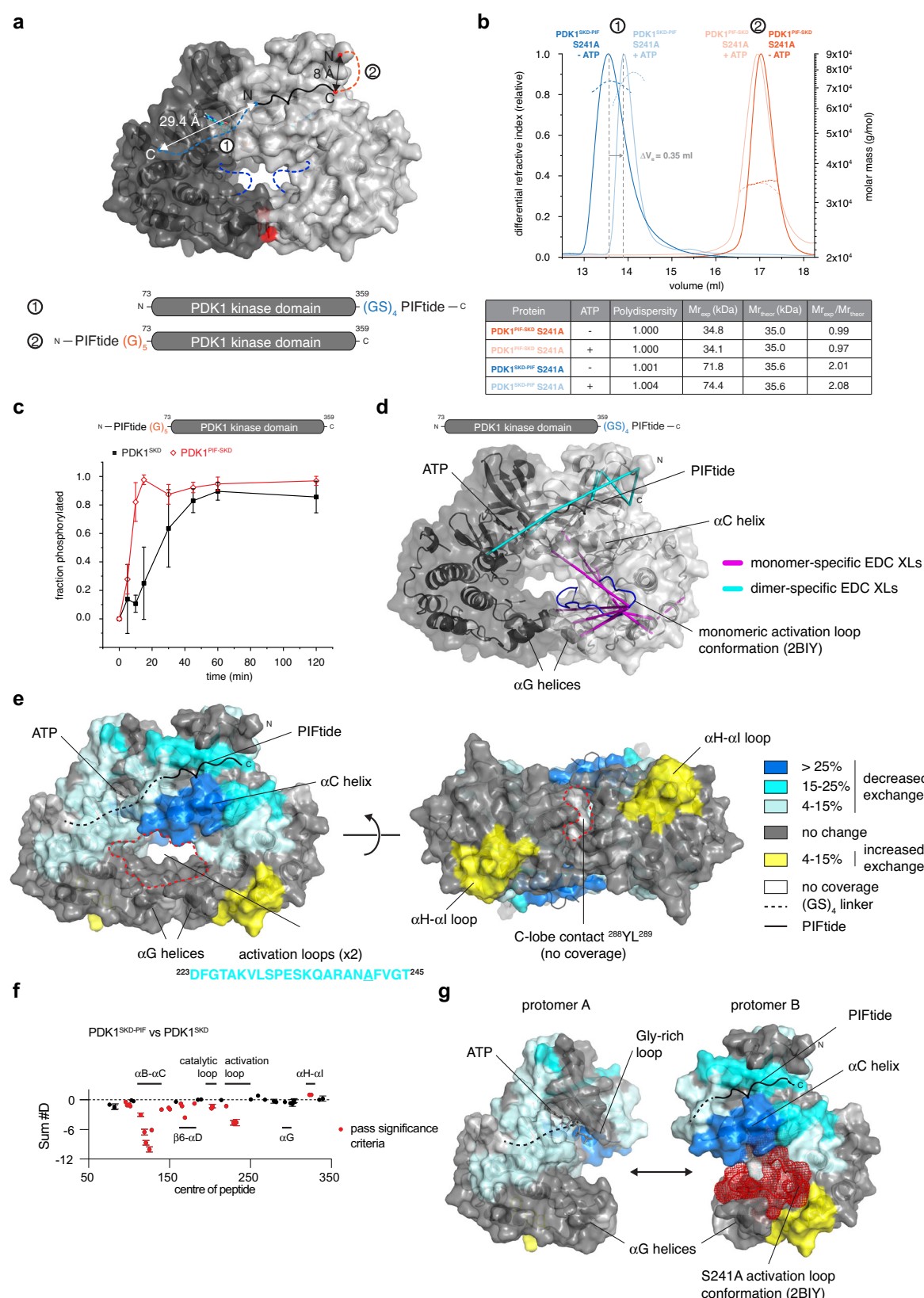

questioned the source of the HM in the autophosphorylation reaction. Sequence alignment of PDK1 orthologs revealed strong conservation in the flexible linker between the kinase and the PH domains (residues 359–389), which was especially pronounced within chordates (Fig. 3a, Supplementary Fig. 5a). We therefore tested whether this sequence influences autophosphorylation by

comparing the kinetics of PDK1$^{SKD}$ (residues 73–359, encompassing just the kinase domain) with a construct of PDK1 comprising an additional 30 amino acids containing the kinase-PH interdomain linker sequence, which we henceforth refer to as PDK1 long kinase domain (PDK1$^{LKD}$, residues 73–389; Supplementary Fig. 1k, l). PDK1$^{LKD}$ autophosphorylation kinetics were

**Fig. 2 PIF pocket occupancy promotes PDK1 *trans*-autophosphorylation. a** Graphical schematics of PDK1$^{SKD-PIF}$ (indicated with the number 1) and PDK1$^{PIF-SKD}$ (indicated with the number 2) fusion proteins. PIFtide is depicted with a black line, (GS)$_4$ in PDK1$^{SKD-PIF}$ is shown in blue dashed line and G$_5$ linker in PDK1$^{PIF-SKD}$ is shown in orange dashed line. Activation loops that were removed from the structure are shown in dark blue dashed lines. **b** Size exclusion chromatography coupled to multi-angle light scattering (SEC-MALS) of PDK1$^{SKD-PIF}$ S241A run in the presence (light blue) or absence (dark blue) of 1 mM ATP and PDK1$^{PIF-SKD}$ S241A run in the presence (light orange) or absence (dark orange) of 1 mM ATP. PDK1$^{SKD-PIF}$ constructs (blue) and PDK1$^{PIF-SKD}$ constructs (orange) are indicated with the numbers 1 and 2, respectively, above the chromatograms. The table summarizes protein polydispersity, theoretical (Mr$_{theor}$) and experimentally determined (Mr$_{exp}$) molecular weights, as well as the calculated oligomeric state (Mr$_{exp}$/Mr$_{theor}$). **c** Radiometric autophosphorylation assay. PDK1$^{SKD}$ (closed black squares) and PDK1$^{PIF-SKD}$ (open red diamond). $n = 3$ biologically independent experiments. Data are presented as mean values ± S.D. **d** Crosslinking mass spectrometry analysis of PDK1$^{SKD-PIF}$ S241A in the presence of 1 mM ATP. EDC crosslinked peptides are mapped on the model of the kinase domain dimer. Monomer-specific crosslinks are shown in purple, dimer-specific crosslinks are shown in cyan. The activation loop is highlighted in blue. **e** Comparison of deuterium incorporation in the PDK1 monomer (PDK1$^{SKD}$) and dimer (PDK1$^{SKD-PIF}$). PIFtide is depicted with a solid black line, the (GS)$_4$ linker in dashed line. The sequence below the model (in cyan) is the sequence of the activation loop that was removed for modeling. Color code for the magnitude of changes in deuterium incorporation is shown to the right. Source data are provided as a Source Data file. **f** Plot of differences in deuterium incorporation between PDK1$^{SKD}$ (reference) and PDK1$^{SKD-PIF}$ (reference). Changes in deuterium incorporation are plotted against the center of each peptide. Regions of protection in PDK1$^{SKD-PIF}$ are indicated below the plot and correspond to those mapped in Fig. 2e. Red data points indicate increases or decreases in exchange that passed the three significance criteria. $n = 3$ biologically independent experiments. Data are presented as mean values ± S.D. **g** The conformation of the phosphorylated activation loop of PDK1 (PDB 2BIY) is shown in red mesh representation on protomer B.

dramatically faster than PDK1$^{SKD}$ (Fig. 3b), indicating that the 30 amino acids C-terminal to the kinase domain promote autophosphorylation. However, the linker did not affect the oligomeric state of PDK1$^{LKD}$S241A (Supplementary Fig. 1m), which remained monomeric at 5 μM even in the presence of 1 mM ATP (Supplementary Fig. 5b).

Despite PDK1$^{LKD}$ remaining monomeric, the faster autophosphorylation kinetics indicated additional interactions in the context of transient kinase domain dimerization. To probe these interactions, we subjected PDK1$^{LKD}$ to crosslinking with the homobifunctional crosslinker bis(sulfosuccinimidyl)suberate (BS$^3$). Bands corresponding to monomeric and dimeric crosslinked PDK1$^{LKD}$ (Supplementary Fig. 5c) were excised from the gel and analyzed by mass spectrometry. Crosslinks were mapped onto the model of the kinase domain dimer (Fig. 3c, Supplementary Table 4). Dimer-specific intermolecular crosslinks (cyan) are compatible with the model, while monomer-specific intramolecular crosslinks (purple) are consistent with conformational changes in the N-lobe upon dimerization, which presumably drives autophosphorylation.

Since PDK1$^{PIF-SKD}$ and PDK1$^{LKD}$ both resulted in accelerated PDK1 autophosphorylation kinetics, we hypothesized that the kinase-PH linker might contain a previously unidentified HM that could bind into the hydrophobic pocket of the opposing protomer upon transient dimerization. We reasoned, that, if this was the case, a fusion of PIFtide to the N-terminus of PDK1$^{LKD}$ (PDK1$^{PIF-LKD}$) would compete with the linker for binding to the hydrophobic pocket, thereby inhibiting stimulation of PDK1 autophosphorylation by the linker. To test this hypothesis, we compared the autophosphorylation kinetics of PDK1$^{PIF-LKD}$ (Supplementary Fig. 1n) to PDK1$^{LKD}$. For this and subsequent kinase assays, it was necessary to adjust the reaction conditions in order to slow the rate of PDK1$^{LKD}$ autophosphorylation sufficiently to obtain high-quality kinetic data. This was achieved by reducing the ATP concentration from 200 to 50 μM and PDK1 concentration from 5 to 1 μM. PDK1$^{PIF-LKD}$ exhibited reduced autophosphorylation kinetics (Fig. 3d), suggesting that the PIFtide sequence competes with the kinase-PH linker for binding to the hydrophobic pocket and indicating that PDK1 encodes a bona fide HM in its linker.

We next sought to identify a putative HM sequence in the linker. The linker sequence contains several patches of hydrophobic residues, but only one corresponds to a typical HM motif ($^{383}$Phe-Gly-Cys-Met$^{386}$) and is capable of reaching into the hydrophobic pocket of the opposing protomer in our in silico model. The sequence of this motif aligns favorably with the

structurally validated HMs of 57 out of 62 other AGC kinases (Supplementary Fig. 5d). Mutation of both of these hydrophobic residues to alanines (PDK1$^{LKD}$ F383A/M386A, Supplementary Fig. 1o) completely abrogated the effect of the linker (Fig. 3e), confirming that the linker contains a HM that binds to the hydrophobic pocket and acts as an allosteric modulator.

However, the increased autophosphorylation kinetics of PDK1$^{LKD}$ compared to PDK1$^{PIF-SKD}$ (Fig. 3d) suggested that the HM is not the only part of the linker that promotes autophosphorylation. A conserved Asn-Phe-Asp (NFD) motif in the C-terminal tail of AGC kinases PKC iota and PKA has been shown to be important for the binding of ATP and consequently, for kinase activity[40,41]. Within the PDK1 linker sequence, we observed an Asn-Tyr-Asp (NYD) motif five residues N-terminal to the HM. Mutation of Y376 to alanine (PDK1$^{LKD}$ Y376A, Supplementary Fig. 2a) resulted in slower autophosphorylation kinetics than the wild-type long kinase domain (PDK1$^{LKD}$) (Fig. 3f) indicating that the NYD motif also promotes PDK1 autophosphorylation.

Taken together, these results suggest that the linker between the kinase and PH domains encodes previously unidentified, but equivalent, NFD and HM motifs found in the C-terminal tail of other AGC kinases. We conclude that PDK1 autophosphorylation is allosterically regulated by its kinase-PH interdomain linker in a mechanism analogous to a domain-swap, but of a necessarily transient nature.

**PDK1 is autoinhibited by its PH domain**. A C-terminal membrane-binding PH domain that binds to PI(3,4)P$_2$ and PIP$_3$ with high affinity allows PDK1 to be targeted to membranes in response to growth factor signaling[8]. PI3K-dependent plasma membrane translocation and a consequent increase in PDK1 dimerization have previously been observed in cells[12]. Based on this, we hypothesized that PDK1 autoactivation might be restricted to PIP$_3$ and PI(3,4)P$_2$-containing membranes. Such a mechanism of membrane-restricted kinase activity has previously been observed in the related AGC kinases Akt and Sgk3, which contain a PIP$_3$- and PI(3,4)P$_2$-binding PH domain or PI3P-binding PX domain, respectively[42–45]. In its inactive state, the catalytic cleft of Akt is occluded by its PH domain, an autoinhibitory interaction that is relieved by PIP$_3$ or PI(3,4)P$_2$ binding[43,45,46]. We therefore hypothesized that PDK1 activation might be regulated in a similar way.

To test whether the PH domain inhibits PDK1 autophosphorylation, we compared the autophosphorylation kinetics of

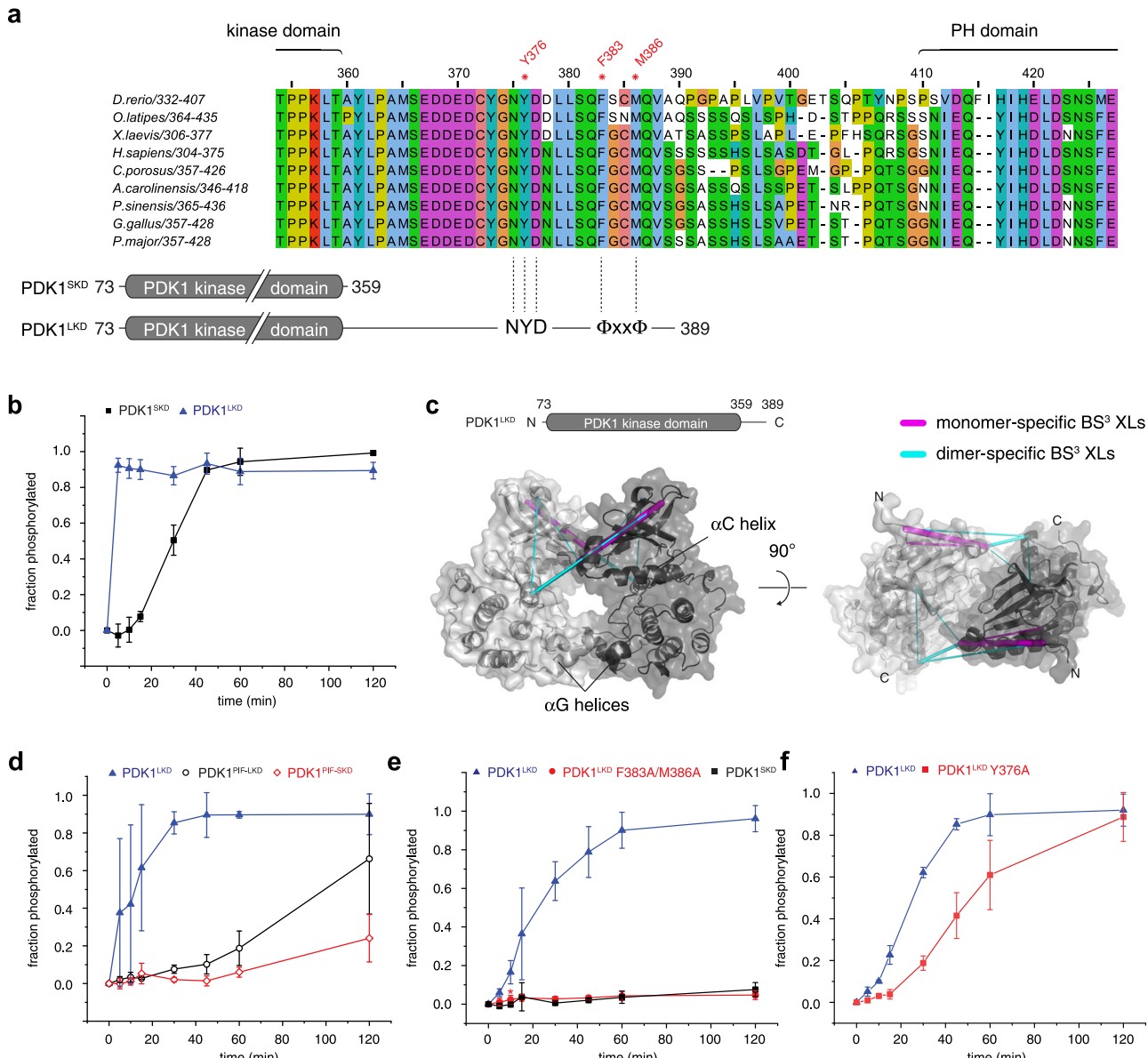

**Fig. 3 A hydrophobic motif in PDK1 drives *trans*-autophosphorylation. a** Alignment of the kinase-PH interdomain linker of chordate PDK1 orthologs. Below: schematic of the PDK1$^{SKD}$ and PDK1$^{LKD}$ constructs with relative positions of the 'NYD' and hydrophobic (ΦxxΦ) motifs. **b** Radiometric autophosphorylation assay. PDK1$^{SKD}$ (closed black squares) and PDK1$^{LKD}$ (closed blue triangles). $n = 3$ biologically independent experiments. Data are presented as mean values ± S.D. **c** Crosslinking mass spectrometry analysis of PDK1$^{LKD}$ S241A in the presence of 1 mM ATP. BS3-crosslinked peptides are mapped on the model of the kinase domain dimer. Monomer-specific crosslinks are shown in purple, dimer-specific crosslinks are shown in cyan. **d** Radiometric autophosphorylation assay. PDK1$^{LKD}$ (closed blue triangles), PDK1$^{PIF-LKD}$ (open black circles), and PDK1$^{PIF-SKD}$ (open red diamonds). $n = 3$ biologically independent experiments. Data are presented as mean values ± S.D. **e** Radiometric autophosphorylation assay. PDK1$^{LKD}$ (closed blue triangles), PDK1$^{LKD}$ F383A/M386A (closed red circles), and PDK1$^{SKD}$ (closed black squares). $n = 3$ biologically independent experiments. Data are presented as mean values ± S.D. (red asterisk indicates removal of one outlier). **f** Radiometric autophosphorylation assay. PDK1$^{LKD}$ (closed blue triangles) and PDK1$^{LKD}$ Y376A (closed red squares). $n = 3$ biologically independent experiments. Data are presented as mean values ± S.D.

near-full-length PDK1 containing its PH domain (residues 73–560; PDK1$^{FL}$, Supplementary Fig. 2b, c) to PDK1$^{LKD}$ and PDK1$^{SKD}$ (Fig. 4a). The autophosphorylation kinetics of PDK1$^{FL}$ were significantly slower than PDK1$^{LKD}$, suggesting that the PH domain exerts an inhibitory effect on the kinase domain. It is worth noting that the inhibitory effect of the PH domain could only be observed by comparing PDK1$^{FL}$ with PDK1$^{LKD}$, since PDK1$^{SKD}$ does not contain the regulatory NYD or HM motifs provided by the kinase-PH linker. Comparison of PDK1$^{SKD}$ with PDK1$^{FL}$ would lead to the erroneous conclusion that the PH domain activates PDK1.

Such an inhibitory interaction between the PH and the kinase domain should be reflected in the overall dimensions of PDK1. We therefore analyzed the conformation of PDK1$^{FL}$ in solution by small-angle X-ray scattering (SAXS) (Fig. 4b-e). We compared PDK1$^{FL}$ to the previously determined dimensions of Akt, since both proteins have almost identical molecular weights (57 and 56 kDa, respectively) and domain compositions. The calculated radius of gyration ($R_g = 29.3$ Å) and maximum dimension ($D_{max} = 82.9$ Å) of PDK1$^{FL}$ are closer to the ones obtained for Akt in the autoinhibited state ($R_g = 27$ Å and $D_{max} = 97$ Å) than for an interface mutant of Akt that adopts an open, PIP$_3$-independent, conformation ($R_g = 32$ Å;

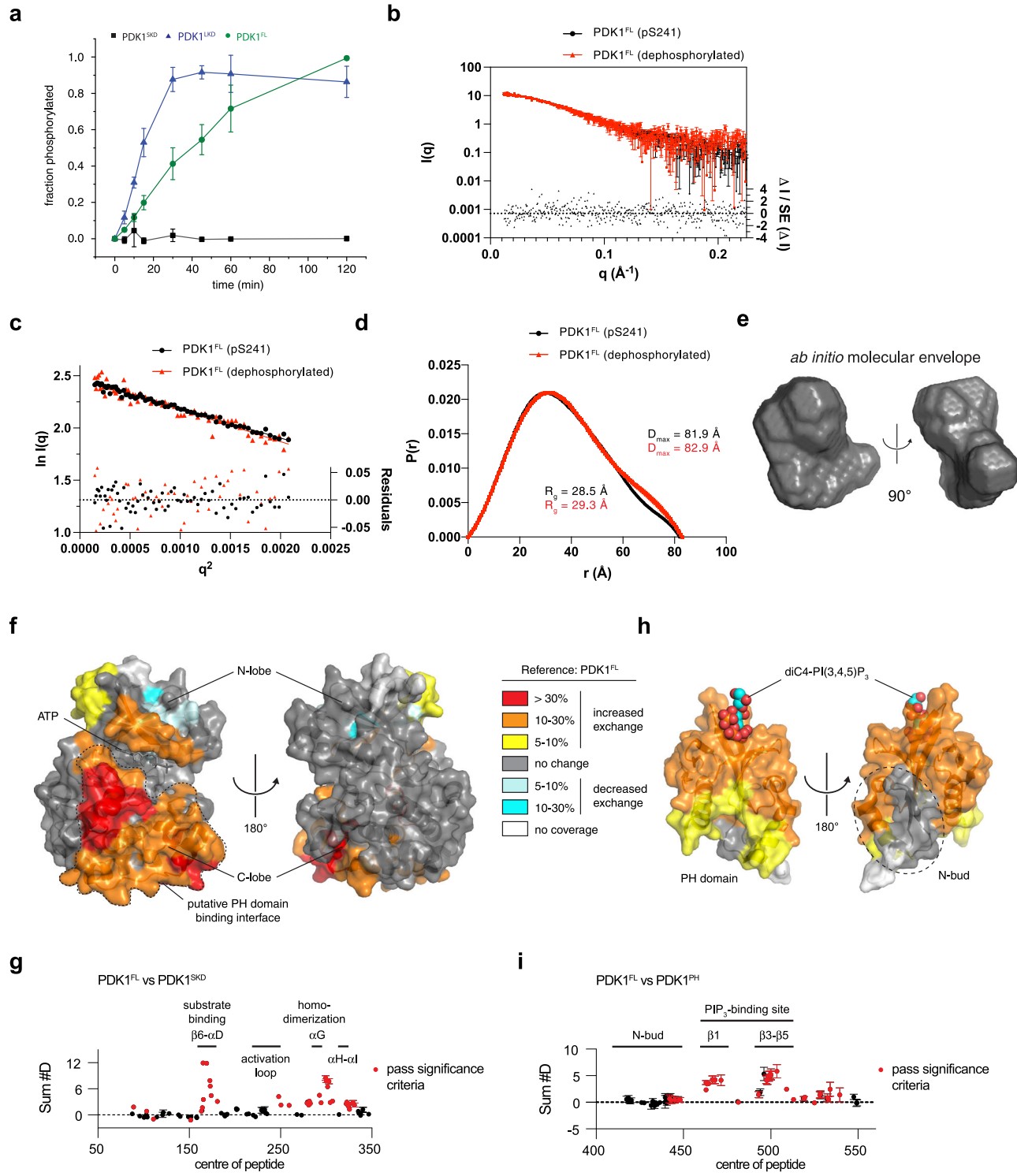

$D_{max} = 122$ Å[43]. Furthermore, the overall dimensions of PDK1[FL] were not affected by activation loop (S241) phosphorylation, demonstrating that PDK1 exhibits the same, compact conformation when stoichiometrically phosphorylated. These observations imply that PH domain-mediated autoinhibition could restrict both PDK1 *trans*-autophosphorylation and activity against downstream substrates to membranes enriched in either PIP$_3$ or PI(3,4)P$_2$.

We next sought to investigate the mechanism and structural basis of PH domain-mediated autoinhibition by HDX-MS. We compared the deuterium incorporation kinetics for the kinase (PDK1[SKD]) and PH (PDK1[PH], Supplementary Fig. 2d) domains of PDK1 with full-length PDK1 (PDK1[FL]) in pairwise HDX-MS experiments. Comparison of PDK1[FL] with PDK1[SKD] revealed a large area of increased exchange on one side of the kinase domain in the absence of the PH domain (Fig. 4f, g). The putative PH domain binding interface covers the whole of the catalytic cleft, including the substrate-binding helix αD, as well as the αG-containing dimerization surface, suggesting that the PH domain mediates PDK1 autoinhibition by impairing kinase

**Fig. 4 PDK1 is autoinhibited by its PH domain. a** Radiometric autophosphorylation assay. PDK1$^{SKD}$ (closed black squares), PDK1$^{LKD}$ (closed blue triangles), and PDK1$^{FL}$ (closed green circles). $n = 3$ biologically independent experiments. Data are presented as mean values ± S.D. **b** Small-angle X-ray scattering (SAXS) curves for S241-monophosphorylated (black) and dephosphorylated (red) PDK1$^{FL}$ in solution. Normalized residuals ($\Delta I/S.E.(\Delta I)$) comparing the scattering curves of S241-monophosphorylated and dephosphorylated PDK1$^{FL}$ are plotted below. **c** Guinier plot of the solution scattering data for S241 monophosphorylated (black) and dephosphorylated (red) PDK1$^{FL}$. The interval of $0.32 < qRg < 1.29$ was used to determine the radius of gyration (Rg) by linear regression of ln I(q) versus q2. Residuals for the Guinier fits of S241-monophosphorylated (black) and dephosphorylated (red) PDK1$^{FL}$ are plotted below. **d** Pair distribution functions for S241-monophosphorylated (black) and dephosphorylated PDK1$^{FL}$ (red) with Rg and maximum particle dimension (Dmax) derived from it. **e** Ab initio calculation of the molecular envelope of PDK1$^{FL}$. **f** Comparison of deuterium incorporation between PDK1$^{FL}$ and PDK1$^{SKD}$ in solution. Peptides showing significant deuterium exchange differences were mapped on the PDK1 kinase domain structure (pdb 2BIY). Color code for the magnitude of changes in deuterium incorporation is shown on the right. Source data are provided as a Source Data file. **g** Plot of differences in deuterium incorporation between PDK1$^{FL}$ (reference) and PDK1$^{SKD}$. Changes in deuterium incorporation are plotted against the center of each peptide. Regions of exposure in PDK1$^{SKD-PIF}$ are indicated above the plot and correspond to those mapped in Fig. 4f. Red data points indicate increases or decreases in exchange that passed the three significance criteria. $n = 3$ biologically independent experiments. Data are presented as mean values ± S.D. **h** Comparison of deuterium incorporation between PDK1$^{FL}$ and PDK1$^{PH}$. Peptides showing significant deuterium exchange differences were mapped on the PDK1 PH domain structure (PDB: 1w1d). Color coding is the same as in panel **f**. Source data are provided as a Source Data file. **i** Plot of differences in deuterium incorporation between PDK1$^{FL}$ (reference) and PDK1$^{PH}$. Changes in deuterium incorporation are plotted against the center of each peptide. Regions of exposure in PDK1$^{PH}$ are indicated above the plot and correspond to those mapped in Fig. 4b. Red data points indicate increases or decreases in exchange that passed the three significance criteria. $n = 3$ biologically independent experiments. Data are presented as mean values ± S.D.

domain association and, consequently, *trans*-autophosphorylation, as well as downstream substrate engagement. To further confirm the interaction between the two domains, we compared PDK1$^{FL}$ to the isolated PH domain (PDK1$^{PH}$). In the absence of the kinase domain, a large part of the PH domain, including the PIP$_3$ binding pocket, shows increased H/D exchange (Fig. 4h, i), suggesting that the PH domain engages in an intramolecular interaction with the kinase domain in full-length PDK1. This surface is highly conserved (Supplementary Fig. 6a) and covers the positively charged phosphoinositide-binding pocket (Supplementary Fig. 6b), but does not include the highly divergent N-bud extension that is peculiar to the PH domain of PDK1 (Supplementary Fig. 6c). In conclusion, PDK1 exhibits a compact, autoinhibited conformation in which its membrane-binding PH domain inhibits both autophosphorylation and blocks substrate binding to the catalytic cleft.

**PIP$_3$ drives cooperative, switch-like PDK1 activation.** PH domain-mediated autoinhibition of PDK1 implies that PIP$_3$ should activate its kinase activity. To probe this directly, we employed activated PDK1$^{FL}$, stoichiometrically phosphorylated on Ser241, in a substrate phosphorylation assay. To simplify interpretation of the results, we used the well-characterized Akt substrate peptide Crosstide, which contains a canonical AGC kinase recognition motif. PDK1 exhibited a five-fold higher initial reaction rate in the presence of liposomes containing 5 mol % PIP$_3$ compared to those containing no PIP$_3$ (Fig. 5a). PDK1 exhibited near-complete binding to PIP$_3$-containing liposomes in this assay (Fig. 5b). The activation of phosphorylated PDK1 by PIP$_3$ is consistent with the absence of conformational changes induced by phosphorylation alone, as observed by SAXS (Fig. 4b-e).

We observed in the HDX-MS data that the surface of the PH domain including the phosphoinositide-binding pocket exhibited lower rates of hydrogen-deuterium exchange in full-length PDK1 compared to the isolated PH domain (Supplementary Fig. 7a-f). This implied that the binding pocket may be sequestered in the autoinhibitory interface between the kinase and PH domains. This property of lipid-regulated kinases has previously been observed for Akt[42], Sgk3[44], PKC[47,48], and PKD[49]. Since the PH domain sequesters the dimerization surface of the kinase domain, we also hypothesized that activation of PDK1 by PIP$_3$ may be highly cooperative, involving kinase domain, NYD, and hydrophobic motif interactions that are only permissible upon the relief of autoinhibition (Fig. 5c). To test both of these hypotheses, we performed a liposome pelleting assay in which we compared the binding affinities of full-length PDK1 with its isolated PH domain. We observed that the PH domain bound to

PIP$_3$-containing liposomes with approximately two-fold higher affinity than full-length PDK1 (Fig. 5d). The binding of the isolated PH domain could not be fit with a one-site binding model (Supplementary Fig. 8a, b), but exhibited positive cooperativity (Hill coefficient $= 1.91 \pm 0.44$), consistent with dimerization of the isolated PH domain on the membrane, as has previously been reported[14]. However, PDK1$^{FL}$ exhibited even stronger positive cooperativity, with a Hill coefficient of $6.01 \pm 0.42$, consistent with additional interactions that stabilize the PIP$_3$-bound conformation, including kinase domain dimerization, NYD motif interactions, and HM exchange.

**Discussion**

Our work describes a molecular mechanism of PDK1 activation. In contrast to previous assumptions that PDK1 is constitutively active in cells, we propose a model in which PDK1 autoactivation and subsequent substrate phosphorylation are restricted to PI(3,4)P$_2$ and PIP$_3$-containing membranes (Fig. 6). Prior to pathway activation by insulin or growth factors, PDK1 is present in the cytoplasm in an autoinhibited conformation in which the substrate-binding cleft and the dimerization surface of the kinase domain, as well as the lipid-binding site of the PH domain, are sequestered in an intramolecular interaction. Upon PI(3,4)P$_3$ or PI(3,4)P$_2$ production in the membrane, PDK1 is recruited via its PH domain, which binds specifically to either lipid, thereby breaking the autoinhibitory interaction. Endogenous PDK1 concentrations have been reported to be ~30 nM[20], a concentration at which we do not observe any dimer formation in vitro. Lipid-mediated recruitment to the membrane results in the local concentration of PDK1, which presumably drives kinase domain dimerization. The linker-swap ensures the specificity of PDK1 dimerization and allosterically promotes *trans*-autophosphorylation via a previously uncharacterized HM. Upon autophosphorylation, the kinase domains of PDK1 dissociate to permit downstream substrate phosphorylation. Whether autophosphorylation drives the dissociation of the kinase domains or dimerization is sufficiently transient that a dedicated mechanism of dissociation is not required, however, remains an open question.

PDK1 dimerization and *trans*-autophosphorylation have been previously reported[10–12]. Masters et al, however, have proposed a model of the PDK1 kinase domain dimer[12] that is quite different from the model proposed here. Our biochemical and structural modeling shows that the HM we have identified in the kinase-PH interdomain linker can reach into the hydrophobic pocket of the other protomer in the dimer. In the model proposed by

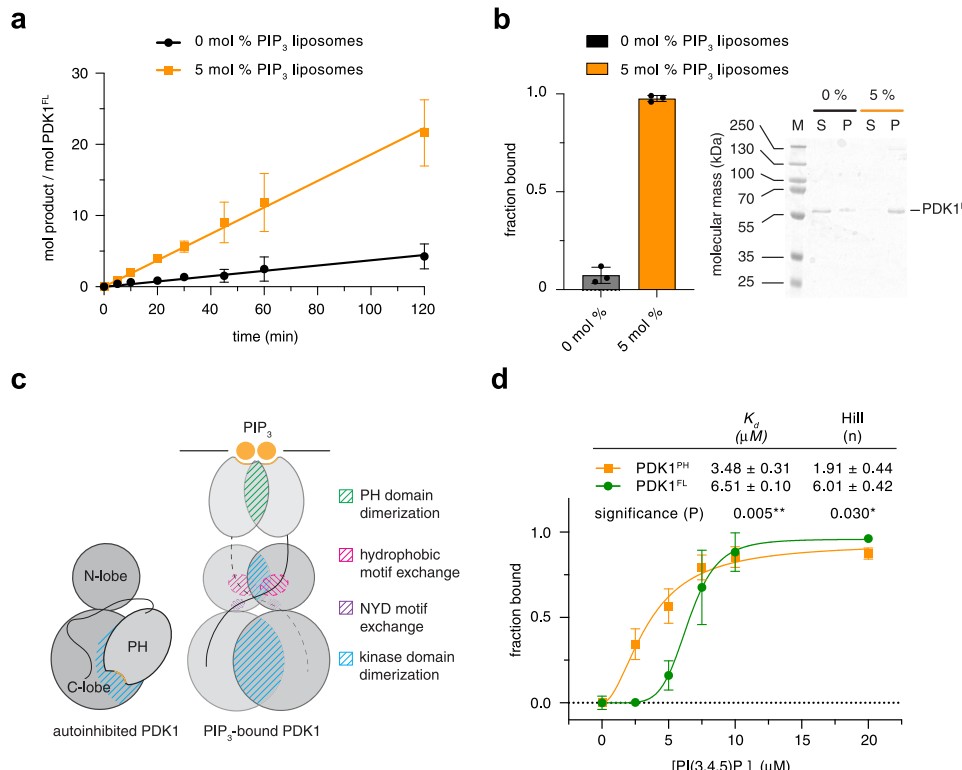

**Fig. 5 PIP₃ drives cooperative, switch-like PDK1 activation. a** Radiometric substrate (Crosstide) phosphorylation assay. Initial PDK1 reaction velocities in the presence of liposomes containing 0 mol % or 5 mol % PIP₃ were determined by linear regression of data points up to 120 min in which substrate consumption was no more than 2.5%. $n = 3$ biologically independent experiments. Data are presented as mean values ± S.D. **b** Binding of PDK1$^{FL}$ to liposomes containing 0 or 5 mol % PIP₃ under kinase assay reaction conditions. $n = 3$ biologically independent experiments. Data are presented as mean values ± S.D. **c** Schematic of the conformational changes predicted upon PIP₃ binding. **d** Binding of PDK1$^{FL}$ and PDK1$^{PH}$ to 0–2 mol % PIP₃ liposomes. $n = 3$ biologically independent experiments. Data are presented as mean values ± S.D. Data were fit with a Hill-Langmuir isotherm in which no parameters were constrained. The equilibrium dissociation constants derived from curve fitting, together with the respective Hill coefficients, are reported in the table (inset). The differences in both Kd and n were determined to be significant by a two-tailed $t$-test ($P = 0.005$ for the difference in Kd and $P = 0.030$ for the difference in $n$).

Masters et al., the αG helix of one protomer interacts with the hydrophobic pocket of the other protomer. The hydrophobic pockets in this dimer are therefore oriented in a way that is incompatible with the binding of the HM from the opposing protomer. Although our structural model is derived from in silico docking of an existing crystal structure, evidence for this mode of kinase domain dimerization is supported by crystal structures of MEK1 and MEK2 homodimers as well as the BRAF:MEK1 heterodimer, all of which are arranged in a face-to-face arrangement that is superimposable with our model of PDK1. Furthermore, the relevance of αG-mediated heterodimerization in B-Raf:MEK signaling is well established[50,51]. αG helix-mediated homodimerization of PDK1 is, however, dependent on PIP₃ or PI(3,4)P₂, since the PH domain obscures the dimerization surface of the kinase domain in the absence of lipids. The same surface of other kinases has been observed to mediate myriad protein-protein interactions that control (auto)inhibition[43,46,52,53], substrate docking[54], recruitment of phosphatases[55], and homo- and heterodimerization-driven *trans*-autophosphorylation[50].

The PH domain has previously been proposed to regulate PDK1 kinase domain dimerization[11,12,14]. These studies propose a model in which PDK1 forms an autoinhibitory dimer in the cytoplasm that is dissociated upon *trans*-autophosphorylation of T513 in the PH domain following the binding of PDK1 to PIP₃ or PI(3,4)P₂ in the membrane. It is unclear, however, what would drive PDK1 dimerization in the cytoplasm considering low

nanomolar PDK1 expression levels and the fact that dimerization cannot be observed up to 2 μM in vitro. Moreover, we did not observe any T513 phosphorylation in full-length PDK1 following stoichiometric activation loop phosphorylation (Supplementary Fig. 2c). Our HDX-MS analysis, however, revealed that T513 is located within the intramolecular interface between the PH and kinase domains. Comparison of the rates of deuterium incorporation for the peptide comprising residues 511–515 of PDK1 shows that the rate of deuterium incorporation is significantly higher in the absence of the kinase domain (Supplementary Fig. 7f). This was, in fact, the largest difference in deuterium incorporation observed over the whole protein. We therefore hypothesize that the insulin- and growth-factor-independent activity of PDK1 T513E previously observed in cells[12,56] may be due to disruption of the autoinhibitory interaction between the kinase and PH domains. In this respect, it is also worth noting that T513 is invariant in chordate PDK1 orthologs. However, though we doubt the physiological role of T513 phosphorylation, PH domain-mediated dimerization of PDK1 on membranes[14] is very much substantiated by the cooperative binding of the PH domain to PIP₃-containing liposomes. This mechanism enhances the specificity of PDK1 dimerization and, together with intramolecular sequestration of both the PIP₃ binding site and the dimerization surfaces of the PH and kinase domains, ensures high positive cooperativity in PIP₃ sensing, leading to switch-like behavior in PDK1 activation.

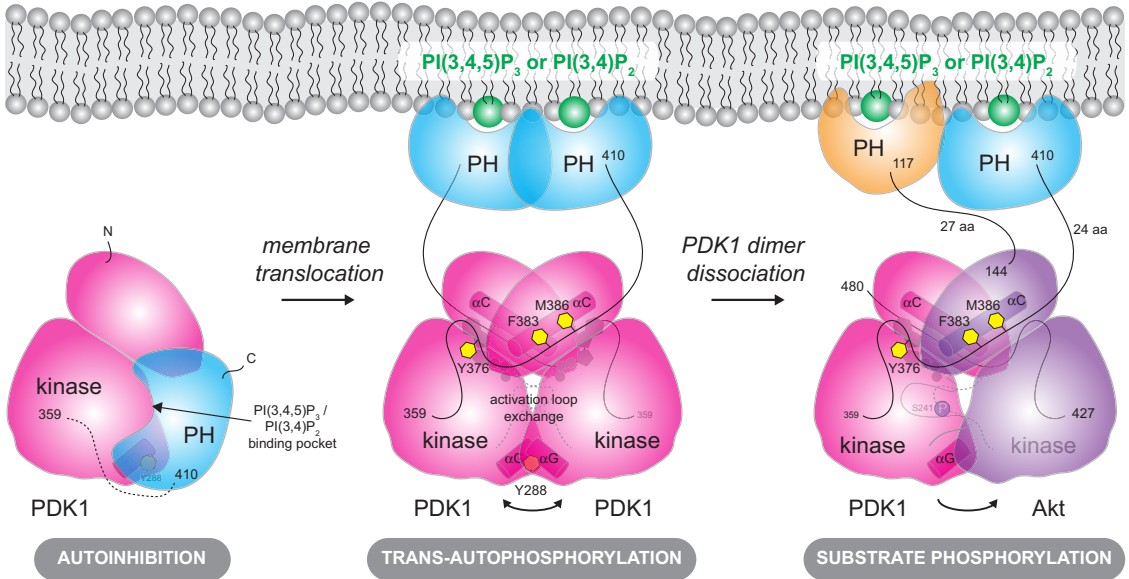

**Fig. 6 Model of phosphoinositide-driven PDK1 activation.** Cytosolic PDK1 is inhibited by its PH domain. PIP$_3$ binding relieves autoinhibition by displacing the PH domain from the dimerization surface and catalytic cleft of the kinase domain. Dimerization of the PH domain upon PIP$_3$ binding leads to PDK1 kinase domain dimerization, interdomain linker exchange (which includes the specific binding of NFD and hydrophobic motifs), and *trans*-autophosphorylation of the activation loop on S241. Upon *trans*-autophosphorylation, the phosphorylated kinase domain dimer dissociates, rendering it competent to bind and phosphorylate downstream substrate kinases. The substrate kinase Akt is used to illustrate the final step in PDK1-mediated downstream kinase activation (far right).

Although the affinities of PDK1 for PIP$_3$ and PI(3,4)P$_2$ were previously reported to be 1.6 and 5.2 nM, respectively[8], it should be noted that these experiments were performed with GST-tagged PDK1. As such, the discrepancy between the reported binding affinities and the values of 3.5–6.5 μM that we have measured is likely attributable to an avidity effect that arises from the use of a dimeric fusion tag. Interestingly, the binding affinity of PDK1$^{FL}$ for PIP$_3$-containing liposomes is only marginally higher than the 15–20 μM determined for Akt1 binding to liposomes of identical composition[42].

The PH domain of PDK1 is essential for its kinase activity. Abrogation of PIP$_3$ binding by mutagenesis of arginines 472, 473, and 474 to leucine leads to embryonic lethality in mice as a consequence of the inability of PDK1 to bind PIP$_3$ or PI(3,4)P$_2$[57]. However, the introduction of a single point mutation, K465E, that disrupts PIP$_3$ binding did not completely abrogate PDK1 substrate phosphorylation, despite leading to impaired mouse development[58]. Our analysis of PIP$_3$ binding in full-length PDK1 indicates that the PIP$_3$ binding site is obscured, an observation corroborated by HDX-MS analysis of the intramolecular interface. In fact, peptides containing K465 exhibit significantly increased rates of deuterium incorporation in the absence of the kinase domain (Fig. 4i, Supplementary Fig. 7a, b). These observations suggest that this mutation might also destabilize the autoinhibitory interaction between the PH and the kinase domains, potentially explaining the residual PDK1 activity and viability observed.

We have identified previously uncharacterized NFD and HMs in the interdomain linker of PDK1 that are equivalent to those found in the C-terminal tail of all other AGC kinases and that allosterically regulate PDK1 activity. Previously, PDK1 was thought to lack the canonical C-terminal extension to its kinase domain. While the fact that the HM sequence in PDK1 is only conserved in chordates raises questions about the mechanism of PDK1 activation in evolutionarily older species, we cannot discount that sequence alignments are not powerful enough to identify an analogous sequence in these species. We show that the HM within the interdomain linker binds to the hydrophobic

pocket of PDK1 during *trans*-autophosphorylation. This interaction, while dramatically stimulating autophosphorylation, is necessary of low affinity compared to the high-affinity interaction of PIFtide, since the kinase domain dimer of PDK1 must be capable of dissociation post-activation loop autophosphorylation in order to engage its substrates. This explains why PKD1$^{LKD}$ is monomeric, while our C-terminal PIFtide fusion protein, PDK1$^{SKD-PIF}$, is constitutively dimeric. Mutation of the central residue in the PDK1 hydrophobic pocket (L155E) was previously shown to result in early embryonic lethality in mice[57]. To date, the hydrophobic pocket in PDK1 has been reported to facilitate the interaction with its substrates. However, our findings indicate that the hydrophobic pocket plays an important role in PDK1 autoactivation, which may explain the strong phenotype observed.

The kinase-PH interdomain linker has previously been reported to mediate PDK1 translocation to the nucleus following insulin stimulation[18,59]. The authors of this study identified a putative nuclear export signal (NES) in mouse PDK1 that overlaps with the HM we have identified in this study. However, data from this study also indicates that the nuclear-cytoplasmic distribution of PDK1 is unaltered in cells lacking PTEN, where PI3K signaling is dramatically upregulated. Whilst we cannot discount the presence of a NES in the interdomain linker, the activation of a nuclear pool of PDK1 would require actively signaling pools of PIP$_3$ or PI(3,4)P$_2$. Though some evidence has hinted at their existence[60], further work is undoubtedly required to probe whether the PI3K pathway is physiologically relevant in the nucleus.

A striking difference between the PDK1 interdomain linker and the C-terminal tail of PDK1 substrates is the absence of a phosphorylated turn motif. The turn motif, reported to be co-translationally phosphorylated by mTORC2[61], binds to a conserved basic patch on the surface of the N-lobe (β1 and β2) of the kinase domain, and is important for regulating the stability of both Akt and PKC[62]. In our model of the *trans*-autophosphorylating PDK1 kinase domain dimer, the N-lobe surface to which the phosphorylated turn motif binds in other AGC kinases

forms a part of the dimerization surface. As such, transient heterodimerization of PDK1 with any of its substrates to accomplish their activation loop phosphorylation might be expected to be incompatible with the docking of the turn motif to the basic patch. Intriguingly, evidence that turns motif phosphorylation results in substrate dissociation from PDK1 has previously been proposed for PRK2[38]. However, little is known about the structural details of PDK1 interaction with its substrates. One obvious hypothesis that arises from this study is that the HM of PDK1 may play an important role in substrate phosphorylation by binding to the vacant hydrophobic pocket of its substrate. Such a mechanism, in which the HMs of both the substrate and PDK1 are exchanged, would undoubtedly enhance the specificity of the interaction, with obvious consequences for signal amplification and fidelity. Further work, however, will be required to test this hypothesis.

Regardless of whether PDK1 forms a specific heterodimer with its substrate or not, our data suggest that PDK1 autoactivation and subsequent substrate phosphorylation is dependent on the presence of $PI(3,4)P_2$ or $PIP_3$ in the membrane. Our finding that even S241 monophosphorylated PDK1 adopts an autoinhibited conformation in the absence of $PIP_3$ suggests that PDK1 may be unable to phosphorylate its substrates in the cytosol. Presumably, therefore, all PDK1 substrates must be targeted to $PIP_3$- or $PI(3,4)P_2$- containing membranes in order to interact with active PDK1. Akt co-localization with PDK1 is ensured by its own PH domain that binds to $PI(3,4)P_2$ and $PIP_3$. Sgk3 phosphorylation by PDK1 is likely restricted to endosomes that present both PI3P, required for Sgk3 activation, and $PI(3,4)P_2$ for PDK1 activation[44]. However, the mechanism of phosphorylation of PDK1 substrates that do not contain a membrane-binding domain, such as RSK, S6K, or PRK2, remains to be elucidated. Whether they require a scaffold protein that would target them to the membrane or they rely on a high-affinity interaction between their HM and the hydrophobic pocket of PDK1 for co-localization is not yet known. However, published data suggest that this may be the case, at least for PRK2 and S6K, which respectively encode either a constitutively high-affinity HM or a HM, which exhibits a dramatic increase in affinity for PDK1 upon phosphorylation[32].

Several PDK1 substrates belong to the PI3K (Akt, Sgk) or MAPK pathways (RSK), both of which are frequently dysregulated in human cancers. PDK1 is, therefore, an attractive target for therapeutic intervention. However, the development of PDK1 inhibitors is challenging, since as little as 10% of normal PDK1 expression levels are enough to maintain viability, while the absence of PDK1 is embryonically lethal[1]. So far, no PDK1 inhibitors have progressed into clinical trials. Several inhibitors targeting the hydrophobic pocket of PDK1 have been developed to target substrate-specific inhibition[24,63]. However, our data showing that the hydrophobic pocket of PDK1 is important for PDK1 autoactivation suggests that inhibitor binding to the pocket would impair activation of all PDK1 substrates. Moreover, the hydrophobic pocket is common to all AGC kinases, with obvious potential for pleiotropic, off-target effects. Our finding that PDK1 is autoinhibited by its PH domain opens up new possibilities for the development of drugs with greater specificity. Allosteric inhibitors targeting the autoinhibited conformation of Akt have been shown to be highly specific[64] with anti-tumor activity in patient-derived xenograft models of pancreatic, colon, and endometrial cancer[65,66] and tumor regression and Akt signaling blockade in advanced solid tumors[67]. Given the essential role of PDK1 in signal transduction downstream of growth factors and upstream of Akt, the development of allosteric inhibitors that target PDK1 may have enormous implications in the treatment of cancer.

## Methods

**Protein expression and purification**. PDK1SKD (73–359), PDK1LKD (73–389), N-terminal PIFtide SKD (PDK1PIF-SKD) or LKD (PDK1PIF-LKD) fusion, and C-terminal PIFtide SKD (PDK1SKD-PIF) fusion were cloned into the pFastBac Dual expression vector with an N-terminal GST-tag for expression in baculovirus-infected Sf9 insect cells. PDK1SKD S241A, PDK1SKD Y288A, PDK1SKD Y288E, PDK1LKD F383A/M386A, PDK1LKD Y376A, and PDK1SKD-PIF S241A were generated by site-directed mutagenesis. Near-full-length PDK1 (residues 73–556; PDK1FL) was cloned into a pFastBac Dual expression vector with an N-terminal GST-tag followed by a 3 C cleavage site and a C-terminal His10-Strep II tag preceded by a TEV cleavage site.

All dephosphorylated proteins were purified according to the following protocol. The pellet from 0.5 L of baculovirus-infected Sf9 cells was lysed in 50 ml of lysis buffer: 50 mM Tris pH 7.5, 300 mM NaCl, 2% glycerol, 1 mM TCEP, 0.25% CHAPS, 1 mM PMSF, 1 mM MgCl2, 0.5 μl benzonase (25 U/μl), 1× protease inhibitor cocktail (made in-house: 100 μM bestatin, 14 μM E-64, 10 μM pepstatin A, 1 μM phosphoramidon). The lysate was cleared by centrifugation at 39,000 × g for 30 min and incubated with 3 ml of glutathione sepharose beads (Cytiva) for 2 h at 4 °C. The beads were washed with Buffer A (50 mM Tris pH 7.5, 300 mM NaCl, 2% glycerol, 1 mM TCEP) and Buffer B (50 mM Tris pH 7.5, 500 mM NaCl, 2% glycerol, 1 mM TCEP) and resuspended in Buffer A containing 1 mM MnCl2 and 40 μM of lambda phosphatase (purified in-house). The protein was dephosphorylated on beads overnight at 4 °C and the next day the beads were extensively washed with Buffer A and Buffer B to remove the lambda phosphatase. The protein was cleaved off the beads in Buffer A with 0.7 μM of 3 C protease (generated in-house) for 2 h at RT. FL constructs were cleaved with 3 C and TEV to remove both tags. The cleaved protein was separated from the beads using a gravity column (Bio-Rad), salt concentration was diluted down to 250 mM using QA buffer (50 mM Tris pH 7.5, 1 mM TCEP) and the protein was loaded on a HiTrap Q FF column (Cytiva). Protein was collected in the flow-through, concentrated, and loaded on a HiLoad S200 Increase 10/300 gel filtration column (Cytiva) equilibrated in gel filtration buffer (50 mM Tris pH 7.5, 100 mM NaCl, 2% glycerol, 1 mM TCEP). S241A mutants were purified in the same way with the omission of the lambda phosphatase step.

The PH domain of PDK1 (408-556) was cloned into pGST parallel with an N-terminal GST-tag and transformed into BL21 STAR electrocompetent E. coli cells. Cells were grown in 4 L of LB medium containing 100 μg/ml ampicillin at 37 °C, 200 rpm to an OD = 0.5. They were then induced with 250 μM isopropylthiogalactopyranoside (IPTG) and grown for 20 h at 20 °C. Cells were centrifuged at 4000 × g for 30 min and pellets were lysed in 200 ml of lysis buffer (50 mM Tris pH 7.5, 150 mM NaCl, 10% glucose, 1 mM EDTA, 1 mM EGTA, 1 mM TCEP, 1× protease inhibitor cocktail, 0.5 μl benzonase (25 U/μl), 2 mM MgCl2, 1 mM PMSF, 0.25% CHAPS). The lysate was sonicated and centrifuged at 39 000 g for 30 min and the supernatant was incubated with 7 ml glutathione sepharose beads (Cytiva) for 2 h at 4 °C. The beads were then washed with 500 ml of buffer A (50 mM Tris pH 7.5, 500 mM NaCl, 2.5 mM DTT) and 200 ml of Buffer B (50 mM Tris pH 7.5, 300 mM NaCl, 2.5 mM DTT). The protein was cleaved off the beads with 500 μMof 3 C protease overnight. The protein was concentrated to 5 ml and injected onto a 16/600 Superdex 200 pg column (Cytiva) equilibrated in buffer B.

**Intact mass spectrometry**. Intact protein samples were diluted in $H_2O$ and up to 100 ng protein were loaded on an XBridge Protein BEH C4 column (2.5 μm particle size, dimensions 2.1 × 150 mm; Waters) using a Dionex Ultimate 3000 HPLC system (Thermo Fisher Scientific) with a working temperature of 50 °C, 0.1% formic acid (FA) as solvent A, 80% acetonitrile, 0.08% FA as solvent B. Proteins were separated with a 6 min step gradient from 10 to 80% solvent B at a flow rate of 300 μL/min and analyzed on a Synapt G2-Si coupled via a ZSpray ESI source (Waters). Data were recorded with MassLynx V 4.1 (Waters) and analyzed using the MaxEnt 1 process to reconstruct the uncharged average protein mass.

**Mass spectrometry of crosslinked samples**. Crosslinking of PDK1SKD-PIF S241A with the zero-length crosslinker EDC was performed in 40 mM MES pH 6.5, 100 mM NaCl. Ten micromolar protein was first incubated with 1 mM ATP and 2 mM MgCl2 for 30 min, then mixed with 2 mM EDC, 5 mM Sulfo-NHS, 1 mM ATP, and 2 mM MgCl2 and incubated for 1 h at RT in the dark. The reaction was quenched with 50 mM Tris pH 7.5 and 20 mM beta-mercaptoethanol, 15 min at RT.

Crosslinking of PDK1LKD S241A with the homobifunctional bis(sulfosuccinimidyl) suberate (BS3) crosslinker was performed in 50 mM HEPES pH 7.5, 300 mM NaCl. 60 μM protein was first incubated with 1 mM ATP and 2 mM MgCl2 for 30 min. Then 600 μM mM BS3 was added and the mix was incubated for 1 h at RT in the dark. The reaction was quenched with 70 mM Tris pH 7.5.

For both crosslinking reactions, crosslinked monomer and dimer species were separated by polyacrylamide gel electrophoresis. Gel bands were excised, cut up, and destained with acetonitrile (ACN) and 50 mM ammonium bicarbonate (ABC). Before each of the following reaction steps, gel pieces were washed with 50 mM ABC and dehydrated in 100% ACN in order to facilitate the uptake of the solutions. Disulfide bridges were reduced in 10 mM dithiothreitol in 50 mM ABC for 30 min at 56 °C. Subsequently the free thiols were alkylated with 50 mM

iodoacetamide in 50 mM ABC in the dark (30 min at RT). Proteins were digested with trypsin (Promega) in 50 mM ABC overnight at 37 °C. The reaction was stopped by adding 10 μl of 10% (v/v) formic acid (FA) and peptides were extracted by sonication with 5% FA, 50% ACN. The samples were dried in the vacuum centrifuge and taken up in 0.1% trifluoro acetic acid, 2% ACN. Peptide samples were injected on an Ultimate 3000 RSLC nano-flow chromatography system, set up with a pre-column for sample loading (PepMapAcclaim C18, 2 cm × 0.1 mm, 5 μm). Peptides were separated on the C18 analytical column (PepMapAcclaim C18, 50 cm × 0.75 mm, 2 μm; all HPLC parts Thermo Fisher Scientific) applying a linear gradient from 2 to 40% of solvent B (80% ACN, 0.08% FA; solvent A 0.1% FA) at a flow rate of 230 nl/min over 120 min. EDC crosslinked peptides were analyzed on an Orbitrap Fusion Lumos mass spectrometer (Thermo Fisher Scientific). For the data-dependent mode, survey scans were acquired in the m/z range of 350–1550 at a resolution of 120,000 at 200 $m/z$, with lock mass on. The AGC target value was set to 4E5 with a maximal injection time of 50 ms. The 15 most intense ions were selected within an isolation width of 1.2 Thomson for maximal 200 ms, and then fragmented in the HCD cell with a normalized collision energy of 30%. Spectra were recorded at a target value of 1E5 with a maximal injection time of 200 ms and a resolution of 30,000. Only peptides with an assigned charge state between +3 and +6 were selected for fragmentation, the peptide match and exclude isotope features were enabled and selected precursors were dynamically excluded from repeated sampling for 30 s. BS³ crosslinked peptides were analyzed on a Q Exactive HF-X Orbitrap mass spectrometer (Thermo Fisher Scientific). Survey scans were obtained in a mass range of 300–1700 $m/z$ with lock mass on, at a resolution of 120,000 at 200 m/z and an AGC target value of 3E6. The 10 most intense ions were selected with an isolation width of 1.6 $m/z$, for maximal 250 ms at a target value of 2E5, and then fragmented in the HCD cell by stepping the collision energy from 27 to 33 %. Spectra were recorded at a resolution of 30,000. Peptides with a charge between +3 and +7 were selected for fragmentation, the peptide match feature was set to preferred, the exclude isotope feature was enabled, and selected precursors were dynamically excluded from repeated sampling for 30 s. The MaxQuant software package, version 1.6.0.16 respect. 2.0.3.0[68] was used to identify the most abundant protein hits searching against the sequence of the PDK1 chimera protein, the proteome of Bombyx mori and Spodoptera from uniprot (2019.01 UP000005204_7091_Bombyx_mori_all.fasta, 2019.09/2021.04_Spodoptera.fasta) and common contaminants. Search settings were set to default values. To identify crosslinked peptides, the spectra were searched against the sequences of the top 10 proteins from the MQ search sorted by iBAQ using pLink software v2.3.9[69]. Carbamidomethylation of cysteine was set as fixed, and oxidation of methionine as variable modification. The enzyme specificity was set to tryptic allowing four missed cleavage sites, and EDC/sulfo-NHS and BS³ were specified according to the crosslinking chemistry. Search results were filtered for 1% FDR (false discovery rate) on the PSM level (peptide-spectrum matches) and a maximum precursor mass deviation of 5 ppm. To remove low-quality PSMs, additionally an e-Value cutoff of <0.001 was applied.

**In silico modeling.** The structure of monomeric PDK1 S241A (PDB: 2biy) was modified to remove activation loop residues 223–245, the conformation of which is unknown for a putative dimeric assembly undergoing *trans*-autophosphorylation. The remainder of the model was not modified. Two independent approaches to modeling a homodimer were employed. In the first, the ROSETTA Symmetric Docking protocol[21] was used to model the dimer by imposing C2 symmetry and searching for dyad-symmetric assemblies with the lowest free energy of docking. The best hit from the modeling was further refined using the ROSETTA protocol for local refinement. The final model was then used to search the Protein Data Bank for homologous assemblies and to design interface mutations to test the model. In the second approach, AlphaFold2[22,23] was used to predict the homodimer for residues Q73-T359. Modeling was performed with and without experimental structure input (templates) and the output models compared to ROSETTA by calculating the r.m.s.d. overall $C_\alpha$ atoms for both chains.

**Size exclusion chromatography coupled to multi-angle light scattering (SEC-MALS).** The oligomeric state and polydispersity of purified recombinant proteins were assessed by SEC-MALS. 50 μl of PDK1^SKD S241A (3 mg/ml), PDK1^LKD S241A (5 mg/ml), PDK1^FL S241A (1.5 mg/ml), PDK1^PIF-SKD S241A (5.6 mg/ml), or PDK1^SKD-PIF S241A (4 mg/ml) was injected onto a Superdex 200 10/300 column (Cytiva) operated by a 1260 Infinity HPLC (Agilent Technologies). Light scattering of a 690 nm laser was detected by a MiniDawn Treos (Wyatt) and the refractive index was measured by a Shodex RI-101 (Shodex) detector. The runs were done in 50 mM Tris pH 7.5, 100 mM NaCl, 2% glycerol, 1 mM TCEP, 2 mM MgCl₂, ±1 mM ATP.

**Radiometric kinase assay.** Autophosphorylation and substrate phosphorylation reactions were done using radiolabeled [γ-³²P] ATP (Hartman Analytic). All reactions were performed in the following buffer: 25 mM Tris pH 7.5, 100 mM NaCl, 2% glycerol, 5 mM DTT at 4 °C for the autophosphorylation reactions and at 22 °C for the substrate phosphorylation reaction. In the reactions comparing autophosphorylation kinetics between PDK1^SKD, PDK1^SKD Y288A and PDK1^SKD

Y288E; PDK1^SKD and PDK1^PIF-SKD; PDK1^SKD and PDK1^LKD, 5 μM of dephosphorylated protein, 200 μM of ATP (Promega) spiked 1:10 with [γ-³²P] ATP and 400 μM MgCl₂ were added to the buffer. In the reactions comparing autophosphorylation kinetics between PDK1^LKD, PDK1^PIF-LKD, and PDK1^PIF-SKD; PDK1^LKD, PDK1^FL, and PDK1^SKD; PDK1^LKD and PDK1^LKD F383A/M386A; PDK1^LKD and PDK1^LKD Y376A, 1 μM of dephosphorylated protein, 50 μM ATP (1:10 [γ-³²P] ATP) and 100 μM MgCl₂ were added to the buffer. At every time point (5, 10, 15, 30, and 45 min, 1 and 2 h), 20 μl aliquots were taken from the reaction and mixed with 5 μl of 500 mM EDTA to terminate the reaction. The quenched reactions were spotted on 0.45 μm nitrocellulose membrane (Cytiva). The membrane was washed 4 × 5 min with 50 ml of 75 mM H₃PO₄ to remove non-incorporated ATP. The washed membranes were placed in scintillation tubes containing 5 ml of dH₂O. Cerenkov radiation was measured with a Tri-Carb 4910 TR liquid scintillator (Perkin Elmer). Phosphate incorporation in the assay comparing PDK1^LKD to PDK1^LKD Y376A was quantified by phosphorimaging. The washed membrane was wrapped in Saran film and exposed to a phosphor screen in a cassette overnight. The screen was imaged with an Amersham Typhoon phosphorimager and the radioactive signal was quantified in ImageJ.

In the substrate phosphorylation reaction, 100 nM PDK1^FL, stoichiometrically phosphorylated on S241, was used to phosphorylate 50 μM Crosstide peptide fused to the C-terminus of SUMO-1. Prior to the phosphorylation reaction, PDK1^FL and SUMO-crosstide were incubated for 30 min with liposomes containing 0 or 5 mol % PIP₃. Liposomes were prepared the same way as for the pelleting assay. The reaction was started by adding 1 mM ATP and 2 mM MgCl₂. The enzyme and the substrate were then separated by polyacrylamide gel electrophoresis. The gel was washed 3x5min in water and was dried with a BioRad gel dryer for 30 min. The dried gel was then wrapped in Saran film and exposed to a phosphor screen in a cassette overnight. The screen was imaged with an Amersham Typhoon phosphorimager and the radioactive signal was quantified in ImageJ and normalized to control amounts.

**Hydrogen-deuterium exchange-mass spectrometry**
*Sample preparation.* HDX reactions for PDK1^SKD S241A (monomer) and PDK1^SKD-PIF S241A (dimer) were conducted in a final reaction volume of 10 μL with a final concentration of 6.7 μM and 26.8 μM for PDK1^SKD S241A and PDK1^SKD-PIF S241A, respectively. The reaction was initiated by the addition of 7.0 μL of D2O buffer (20 mM pH7.5 HEPES, 100 mM NaCl, 94% D2O (V/V)) to 3.0 μL of protein solution (final D2O concentration of 66%). The reaction proceeded for 3, 30, 300, or 3000 s at 18 °C, before being quenched with ice-cold acidic quench buffer, resulting in a final concentration of 0.6 M guanidine-HCl and 0.9% formic acid post quench. All conditions were stored at −80 °C, and timepoints were created and run in triplicate.

HDX reactions comparing PDK1^FL and PDK1^SKD were conducted in a final reaction volume of 20 μL with a final protein concentration of 26 μM. The reaction was initiated by the addition of 45 μL of D2O buffer (20 mM pH 7.5 HEPES, 100 mM NaCl, 94% D2O (V/V)) to 5.0 μL of protein solution (final D2O concentration of 89.6%). The reaction proceeded for 3 s at 4 °C and 3, 30, 300, or 3000 s at 18 °C, before being quenched with ice-cold acidic quench buffer, resulting in a final concentration of 0.6 M guanidine-HCl and 0.9% formic acid post quench. All conditions were stored at -80 °C, and timepoints were created and run in triplicate.

HDX reactions comparing PDK1^FL and PDK1^PH were conducted in a final reaction volume of 50 μL with a final protein concentration of 5 μM. The reaction was initiated by the addition of 19 μL of D2O buffer (50 mM pH7.5 HEPES, 100 mM NaCl, 94% D2O (V/V)) to 1.0 μL of protein solution (final D2O concentration of 81.2%). The reaction proceeded for 3 s at 4 °C and 3, 30, 300, or 3000 s at 18 °C, before being quenched with ice-cold acidic quench buffer, resulting in a final concentration of 0.6 M guanidine-HCl and 0.9% formic acid post quench. All conditions and timepoints were created and run in triplicate. Samples were flash-frozen immediately after quenching and stored at −80 °C until injected onto the ultra-performance liquid chromatography (UPLC) system for proteolytic cleavage, peptide separation, and injection onto a QTOF for mass analysis, described below.

*Protein digestion and MS/MS data collection.* Protein samples were rapidly thawed and injected onto an integrated fluidics system containing a HDx-3 PAL liquid handling robot and climate-controlled (2 °C) chromatography system (LEAP Technologies), a Dionex Ultimate 3000 UHPLC system, as well as an Impact HD QTOF Mass spectrometer (Bruker). The protein was run over either one (at 10 °C) or two (at 10 °C and 2 °C) immobilized pepsin columns (Trajan; ProDx protease column, 2.1 × 30 mm PDX.PP01-F32) at 200 μL/min for 3 min. The resulting peptides were collected and desalted on a C18 trap column (Acquity UPLC BEH C18 1.7 mm column (2.1 × 5 mm); Waters 186003975). The trap was subsequently eluted in line with an ACQUITY 1.7 μm particle, 100 × 1 mm² C18 UPLC column (Waters), using a gradient of 3–35% B (Buffer A 0.1% formic acid; Buffer B 100% acetonitrile) over 11 min immediately followed by a gradient of 35–80% over 5 min. Full details of all LC methods can be found at[70]. Mass spectrometry experiments acquired over a mass range from 150 to 2200 m/z using an electrospray ionization source operated at a temperature of 200 C and a spray voltage of 4.5 kV.

*Peptide identification*. Peptides were identified from the non-deuterated samples of PDK1 using data-dependent acquisition following tandem MS/MS experiments (0.5 s precursor scan from 150–2000 *m/z*; twelve 0.25 s fragment scans from 150–2000 *m/z*). MS/MS datasets were analyzed using PEAKS7 (PEAKS), and peptide identification was carried out by using a false discovery-based approach, with a threshold set to 1% using a database of known contaminants found in *Sf9* and *E. coli* cells[71]. The search parameters were set with a precursor tolerance of 20 ppm, fragment mass error 0.02 Da, charge states from 1 to 8, leading to a selection criterion of peptides that had −10logP scores of 31.8, 30.7, 19.7, and 19.1.

*Mass analysis of peptide centroids and measurement of deuterium incorporation*. HD-Examiner Software (Sierra Analytics) was used to automatically calculate the level of deuterium incorporation into each peptide. All peptides were manually inspected for correct charge state, correct retention time, appropriate selection of isotopic distribution, etc. Deuteration levels were calculated using the centroid of the experimental isotope clusters. Results are presented as relative levels of deuterium incorporation, no correction for back exchange, and with the only correction being applied correcting for the deuterium oxide percentage of the buffer utilized in the exchange (66%, 75.5%, 89.6%, and 81.2%) Differences in exchange in a peptide were considered significant if they met all three of the following criteria:: ≥5% change in exchange, ≥0.4 Da difference in exchange, and a two-tailed *t*-test value of *p* < 0.01. The raw HDX data are shown in two different formats. Samples were only compared within a single experiment and were never compared to experiments completed at a different time with a different final D2O level. The data analysis statistics for all HDX-MS experiments are in Supplementary Table 5 according to the guidelines of[72]. The raw deuterium incorporation data for all experiments is available in the source data.

**Small-angle X-ray scattering (SAXS)**. SAXS data for unphosphorylated and S241-monophosphorylated PDK1$^{FL}$ were collected on BM29 at the ESRF, Grenoble, France using an in-line SEC-SAXS setup. Proteins were applied to a Agilent Bio SEC 300 column equilibrated in 20 mM Tris, pH 7.4, 150 mM NaCl, 1 mM DTT, 1 mM EDTA, and 1 % (v/v) glycerol and images were acquired every second for the duration of the size exclusion run. Buffer subtraction was performed by averaging 50 frames on either side of the peak. All subsequent data processing steps were performed using the ATSAS data analysis software 3.9.1. The program DATGNOM[73] was used to generate the pair distribution function [P(r)] for each isoform and to determine $D_{max}$ and $R_g$ from the scattering curves [I(q) vs. q] in an automatic, unbiased manner.

**Preparation of sucrose loaded vesicles (SLVs) and pelleting assay**. Cholesterol, DOPC, DOPS, and DOPE were dissolved in chloroform, and PIP$_3$ has dissolved in a chloroform:methanol:water (1:2:0.8) mixture. To generate SLVs, lipids were mixed in the following molar ratio: 20% cholesterol, 30% DOPC, 15% DOPS, 35% DOPE, and 0–2% PIP$_3$ was added at the expense of DOPC. The lipid mixture was first dried under a nitrogen stream and then rehydrated in 20 mM HEPES pH 7.4, 0.3 M sucrose buffer. Lipid mixtures were then frozen in liquid nitrogen and sonicated at RT. The freeze-thawing steps were repeated four times after which the vesicles were pelleted in a Beckman Coulter Optima MAX-XP Ultracentrifuge using a TLA 100 rotor operated at 108,726 × *g* at 20 °C for 30 min. Liposome pellets were then resuspended in the reaction buffer (50 mM Tris pH 7.5, 150 mM NaCl, 1 mM TCEP) to a final lipid concentration of 1 mM. 2 μM of PDK1$^{FL}$ and 2 μM PDK1$^{PH}$ were mixed 1:1 with liposomes in 50 μl final volume.

For liposome pelleting under kinase assay reaction conditions, 1 μM PDK1$^{FL}$ was mixed 1:1 with liposomes containing either 0 or 5 mol % PIP$_3$ in the presence or absence of 2 mM MgCl$_2$ in the buffer. For both sets of reactions, the mixture was incubated for 30 min and the mixture was incubated for 30 min. After that, the reaction was spun down at 9800 × *g* for 30 min at 20 °C. The supernatant was collected and the pellet was resuspended in the reaction buffer. Equal volumes of supernatant and pellet were loaded on the SDS-PAGE and the fraction of protein-bound to the liposomes was determined by Coomassie densitometry.

**Reporting summary**. Further information on research design is available in the Nature Research Reporting Summary linked to this article.

## Data availability
The data that support this study are available from the corresponding author upon reasonable request. Mass spectrometry proteomics data have been deposited to the ProteomeXchange Consortium via the PRIDE partner repository[74] with the dataset identifier PXD031827. Mass spectrometry HDX-MS proteomics data have been deposited to the ProteomeXchange Consortium via the PRIDE partner repository[74] with the dataset identifier PXD027401. Source data are provided with this paper. HDX-MS data are contained in the source data file. Source data are provided with this paper.

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

## Acknowledgements

We thank Dorothea Anrather and Markus Hartl in the Max Perutz Labs Mass Spectrometry Facility for intact mass, tandem mass spectrometry mapping of post-translational modifications, and crosslinking mass spectrometry analyses, which were performed on instruments of the Vienna BioCenter Core Facilities (VBCF). We also thank Petra Pernot for SAXS data collection on BM29 at the European Synchrotron Radiation Facility (ESRF). This work was supported by the Austrian Science Fund (FWF) in grants P30584, P33066, and W1261 to T.L. J.E.B. is supported by a Michael Smith Foundation for Health Research (MSFHR) Scholar award (17686), and an operating grant from the Cancer Research Society (CRS-24368).

## Author contributions

A.L. purified all proteins and performed all biochemical experiments, with the exception of HDX-MS experiments. T.L. performed in silico modeling. A.L. and T.L. analyzed SAXS data. K.F. carried out all HDX-MS experiments and K.F. and J.B. analyzed the corresponding data. A.L. and T.L. wrote and edited the manuscript. T.L. conceived the project.

## Competing interests

The authors declare no competing interests.
