## [Peer Review File · Nature Communications]

Activation of the essential kinase PDK1 by phosphoinositide-driven trans-autophosphorylationReviewers' Comments:

Reviewer #1:

Remarks to the Author:

The authors employ a multi-pronged, reductionist strategy to dissect the intra- and interdomain interactions responsible for regulation of PDK1 activity. Careful comparisons of various domain truncations and point-mutations reveal several new aspects of PDK1 regulation. By combining crosslinking, HDX-MS, SAXS, and functional assays, the authors develop a model of regulation involving PH-mediated autoinhibition. Auto-activation is proposed to involve de-repression of the PH interaction by PIP3 binding followed by face-to-face dimerization and autophosphorylation in trans. Importantly, the authors identify a motif in the kinase—PH linker that putatively occupies the hydrophobic pocket for allosteric activation of the kinase.

Generally, I am enthusiastic about the mechanistic insights provided by this study. This biochemical investigation provides molecular details that clarify interpretations of the spatiotemporal regulation of PDK1 in cells. The complementary experimental approaches are well conceived and carefully performed. The writing is very clear. These strategies and outcomes would be broadly interesting to the communities focused on signaling mechanisms and drug development. While I agree that the model proposed by the authors is persuasive, I believe that some of the interpretations sprinkled throughout the manuscript are overly adamant. There seems to be room for more nuance here. I have described these points below.

Minor Points:

1. The model proposed in this manuscript focuses a lot on a 'transient dimer' that may involve a 'linker-swap.' Various lines of evidence (e.g., activity assays, mutants, HDX-MS) do convincingly indicate that a face-to-face docking surface is involved in autophosphorylation. However, the writing tends to suggest that a stable complex that the data doesn't directly support. Here are a few examples:

1a. pg 7 "The dimerization of wild-type and mutant kinase domains was evaluated with ... autophosphorylation assay..." This assay convincingly shows that disrupting the surface of this putative docking site impairs autophosphorylation, but it doesn't directly show protein interactions/dimerization.

1b. The only condition that generates a stable dimer is an artificial system here an HM motif is tethered to the terminus. Inclusion of the newly discovered, natural PDK1 motif in the kinase-PH linker does not induce stable dimerization (via SEC-MALS, at least). So saying (pg 12) "previously unidentified HM that could bind into the hydrophobic pocket of the opposing protomer in the dimer" seems to presuppose a stable dimer that is not evident.

1c. Likewise, on pg. 14, "We conclude that PDK1 autophosphorylation is... regulated... in a mechanism analogous to a domain-swap" also leaves the impression of a more stable complex than the evidence warrants.

1d. On pg. 15, "PH domain mediates PDK1 autoinhibition by impairing dimerization..." Here, "dimerization" seems an exaggeration again because PDK1-FL is not a stable dimer.

Importantly, this is not a major disagreement about mechanism. I think it's just semantics. I would prefer that the authors loosen up the "dimerization" language to be more suggestive of a docking interface or surfaces required for productive autophosphorylation substrate recognition. The discussion makes it clear that the authors recognize the importance of a transient recognition surface for autophosphorylation because PDK1 needs to engage with downstream kinases for subsequent phosphorylations (pg 21). But the writing occasionally (e.g., abstract and the examples listed) suggest a more stable architecture than the data warrants.

2. The HDX-MS data analysis and reporting is very well performed, and the results are striking. I have a quibble about an over-simplification in the introduction to the technique (pg 10). HDX rates are sensitive to many factors (e.g., dynamics, secondary structure, etc), including, to some degree,

"solvent accessibility." But I think it's generally agreed that the dominant factors influencing exchange rates involve the stability of main chain H-bonding dynamics. This clarification in the introductory sentence is especially important because the nuances of HDX factors is invoked to explain the lack of G-helix rate perturbation.

3. On a related note, HDX-MS "confirming binding of the PIFtide" (pg 11) is an overstatement. Suppressed exchange rates are consistent with a binding interface, but HDX-MS can't distinguish direct binding from remote, allosteric perturbations or overall changes in the conformational dynamics. I urge more nuance in the explanation of the HDX data here.

4. Because HDX rates are influenced by many factors beyond solvent accessibility, I recommend the substitution of "deprotection" for "exposure" in the HDX figure color code legends.

5. Typo? Figure 1. panel lettering (A-D) appears to omit panel A in the legend. The subsequent Fig. 1 panel letter designations in the legend would then need to be updated.

6. Figure 4. This seems clear enough in the main text of the Results section, but it would be helpful to clarify- in the figure- which protein construct is the "reference" state (i.e., X% protection/deprotection with respect to which state?).

7. Regarding the XL-MS, I see that the dimer- and monomer-specific crosslinks were distinguished by resolving these states on SDS-PAGE. However, can this analysis confirm that the dimer-specific crosslinks are actually intermolecular? Couldn't dimerization induce conformational rearrangements that favor new intramolecular crosslinks? My question here is mostly about the depiction in Fig. 2D and the interpretation "formed by exchange of the C-terminal PIFtide" (pg. 10).

Reviewer #2:

Remarks to the Author:

Activation of PDK1 by PIP3-driven autophosphorylation.

Levina et al, 2022.

This manuscript describes a model of PDK1 auto-activation that involves disengagement of the PH domain (by PIP3-binding) and homo-dimerisation that is driven by face-to-face contacts in the kinase domain and occupation of the PH-kinase linker in a groove on the adjacent protomer. Although there is no experimental structure of the PDK1-PDK1 interaction (affinity too low), the authors use alpha-fold and rosetta to predict the structure of the dimer and then use a number of different biochemical experiments (HDX, SEC-MALS, SAXS, activation assays, cross-linking) to support their model.

Given that PDK1 is a master kinase that activates a myriad of biological pathways, a better understanding of its regulation will have significant impact on our understanding of downstream kinase signalling cascades and this manuscript will be of great interest to those in the field. I recommend publication but just have some minor comments listed below.

General comments:

There appears to be a lot more to the interface than Y288 and the model would be more convincing if further validated by other mutations in this area.

While the biochemical data is strong, the manuscript would benefit from a cellular assay in which the different PDK1 constructs are tested in a more biological context. Are there well-established assays to monitor PDK1 activity in cells that could be used on their mutants?

The authors use SAXS in figure 4. SAXS would be a powerful method to show a change in overall dimensions of the protein upon pip3 binding. Would it be feasible to add the pip3 headgroup (or other soluble pip3) and determine changes in radius of gyration or maximum interatomic distance to show disengagement of the PH domain from the kinase domain?

Much is made of the difference in K_d for FL vs PH binding to pip3 liposomes in Figure 4D, as well as differences in Hill-slope. However, a 2-fold difference in K_D is rarely significant and there are only a few points on these curves and certainly not enough to draw definitive conclusions of Hill-slope (especially for the PH domain, which looks like it would fit equally well to the standard hyperbolic saturation curve). A few more data points in the 0-5 μ M range would be very useful here.

Specific comments:

Figure 2

2B: colour scheme should be changed to be colour-blind friendly. More detailed main text description of which construct (SKD-PIF/PIF-SKD) corresponds to which peak/curve in the figure for clarification.

2E-H: Changes in HDX can be due to dimerization-induced protection from solvent or conformational change or PIF-induced protection from solvent. The authors are perhaps a bit too definite on which of these factors lead to protection of particular residues. Are they using their data to support their model, not their model to support their data? The observable is the rate of deuterium incorporation and the changes in that rate between the two constructs, it would be better to describe the HDX data as "supporting....." rather than "reflecting.....".

The α G helix should be labelled in the figure since it is an important component of the interaction and the fact that there is no change in HDX in this region is more than "surprising" as stated in the text. There does not need to be conformational change to protect deuterium incorporation when the region in question is in the middle of a dimer interface. More effort in the text to explain this unusual result is needed

Figure 4

4A: While I agree with the authors that comparing the phosphorylation kinetics of PDK1-FL and PDK1-SKD is misleading, perhaps a good control would be PDK1-FL F383A/M386A, where the HM has lost its ability to bind the hydrophobic pocket? This should behave like PDK1-SKD, based on figure 3E?

SUMMARY OF MAJOR CHANGES

- The most significant change to the manuscript is Figure 5A-B. While the data presented in the original submission clearly indicated autoinhibition of PDK1 by its PH domain and activation by PIP3, the direct activation of PDK1 by PIP3 was missing. We have now been able to demonstrate this directly in a radiometric substrate phosphorylation assay, which we believe strengthens our conclusions considerably. This experiment also directly addresses reviewer #2's suggestion to employ SAXS to reveal conformational changes upon PIP3 binding. Instead of another indirect way to visualize these changes, we can now directly correlate PIP3 binding with at least 5-fold increased PDK1 activity. We have restructured the old Figure 4 into two Figures now, due to the inclusion of this new data. Figure 4 in the new manuscript now deals exclusively with the structural and mechanistic details of autoinhibition, while Figure 5 deals with the mechanistic details of PIP3-dependent membrane binding and PDK1 activation.
- Reviewer #2 expressed some concerns about the modeling/interpretation of the data presented in Figure 4D-E (now Figure 5C-D). Since the binding model (single-site binding or single-site with Hill coefficient >1) is important to the overall model we present, we have provided two additional Supplementary Figures (Supplementary Figure 8A-B) to illustrate the difference between the two models. An F-test clearly rejects the one-site binding model in favor of a one-site binding with a Hill-coefficient of ~ 2 . Inspection of the residuals of the fits also confirms that the one-site binding model is a poor model of the data (panel B). Since we realized that many readers may share similar concerns, we decided that a Supplementary Figure was important, rather than a private rebuttal figure to the reviewer.
- We have added in a new piece of experimental data to Figure 3 (Figure 3C, Supplementary Figure 5C). This is a cross-linking mass spectrometry experiment in which we assessed the transient dimerization of our PDK1^{LKD} construct. The cross-links obtained are entirely consistent with the Rosetta and AlphaFold models presented in Figure 1, as well as the biochemistry presented in Figures 2-3. This should allay any concerns that the arrangement of the dimer presented in Figure 2 is an artifact of protein engineering (although neither reviewer expressed these concerns, we felt that this was an important control).

REVIEWER COMMENTS

Reviewer #1 (Remarks to the Author):

The authors employ a multi-pronged, reductionist strategy to dissect the intra- and interdomain interactions responsible for regulation of PDK1 activity. Careful comparisons of various domain truncations and point-mutations reveal several new aspects of PDK1 regulation. By combining crosslinking, HDX-MS, SAXS, and functional assays, the authors

develop a model of regulation involving PH-mediated autoinhibition. Auto-activation is proposed to involve de-repression of the PH interaction by PIP3 binding followed by face-to-face dimerization and autophosphorylation in trans. Importantly, the authors identify a motif in the kinase—PH linker that putatively occupies the hydrophobic pocket for allosteric activation of the kinase.

Generally, I am enthusiastic about the mechanistic insights provided by this study. This biochemical investigation provides molecular details that clarify interpretations of the spatiotemporal regulation of PDK1 in cells. The complementary experimental approaches are well conceived and carefully performed. The writing is very clear. These strategies and outcomes would be broadly interesting to the communities focused on signaling mechanisms and drug development. While I agree that the model proposed by the authors is persuasive, I believe that some of the interpretations sprinkled throughout the manuscript are overly adamant. There seems to be room for more nuance here. I have described these points below.

- We appreciate the reviewer's enthusiasm for our study and have addressed all the individual comments/criticisms below.

Minor Points:

1. The model proposed in this manuscript focuses a lot on a 'transient dimer' that may involve a 'linker-swap.' Various lines of evidence (e.g., activity assays, mutants, HDX-MS) do convincingly indicate that a face-to-face docking surface is involved in autophosphorylation. However, the writing tends to suggest that a stable complex that the data doesn't directly support. Here are a few examples:

1a. pg 7 "The dimerization of wild-type and mutant kinase domains was evaluated with ... autophosphorylation assay..." This assay convincingly shows that disrupting the surface of this putative docking site impairs autophosphorylation, but it doesn't directly show protein interactions/dimerization.

1b. The only condition that generates a stable dimer is an artificial system here an HM motif is tethered to the terminus. Inclusion of the newly discovered, natural PDK1 motif in the kinase-PH linker does not induce stable dimerization (via SEC-MALS, at least). So saying (pg 12) "previously unidentified HM that could bind into the hydrophobic pocket of the opposing protomer in the dimer" seems to presuppose a stable dimer that is not evident.

1c. Likewise, on pg. 14, "We conclude that PDK1 autophosphorylation is... regulated... in a mechanism analogous to a domain-swap" also leaves the impression of a more stable complex than the evidence warrants.

1d. On pg. 15, "PH domain mediates PDK1 autoinhibition by impairing dimerization..." Here, "dimerization" seems an exaggeration again because PDK1-FL is not a stable dimer.

Importantly, this is not a major disagreement about mechanism. I think it's just semantics. I would prefer that the authors loosen up the "dimerization" language to be more suggestive

of a docking interface or surfaces required for productive autophosphorylation substrate recognition. The discussion makes it clear that the authors recognize the importance of a transient recognition surface for autophosphorylation because PDK1 needs to engage with downstream kinases for subsequent phosphorylations (pg 21). But the writing occasionally (e.g., abstract and the examples listed) suggest a more stable architecture than the data warrants.

- Interestingly, this issue of semantics with respect to ‘dimerization’ has been raised by some of our scientific colleagues in-house. We tend to believe that the term ‘dimerization’ only implies a specific homo- or hetero- interaction, but does not comment on its stability/affinity. However, we also appreciate that others interpret the term differently. In fact, as the reviewer notes, it would be absurd for PDK1 to form a stable dimer, since this would prohibit downstream signaling. We have revised our language to always explicitly reference a ‘transient dimer’ or ‘transient association’. We are reluctant to replace the concept of ‘dimerization’ with ‘interaction’ as this more often than not has the feel (again, this is a semantic issue) of something less specific than would clearly be required for trans-autophosphorylation. We hope that this intermediate path is acceptable to the reviewer.

2. The HDX-MS data analysis and reporting is very well performed, and the results are striking. I have a quibble about an over-simplification in the introduction to the technique (pg 10). HDX rates are sensitive to many factors (e.g., dynamics, secondary structure, etc), including, to some degree, “solvent accessibility.” But I think it’s generally agreed that the dominant factors influencing exchange rates involve the stability of main chain H-bonding dynamics. This clarification in the introductory sentence is especially important because the nuances of HDX factors is invoked to explain the lack of G-helix rate perturbation.

- We agree with the reviewer that we did not convey very accurately what HDX-MS reports on and that a clarification is necessary particularly with respect to the arguments made about the lack of alphaG helix rate perturbation. We have therefore revised the text accordingly:

“HDX-MS reports the exchange rate of amide hydrogens, and as the primary determinant of amide exchange is the involvement in secondary structure, it acts as a probe to measure changes in secondary structure dynamics (39).”

3. On a related note, HDX-MS “confirming binding of the PIFtide” (pg 11) is an overstatement. Suppressed exchange rates are consistent with a binding interface, but HDX-MS can’t distinguish direct binding from remote, allosteric perturbations or overall changes in the conformational dynamics. I urge more nuance in the explanation of the HDX data here.

- The reviewer is absolutely correct here. We have modified the language here to more accurately reflect what can be reasonably interpreted from the data:

“In the N-lobe, the largest decrease in exchange is observed around the hydrophobic

pocket (α C and α B helices), consistent with stabilization of the N-lobe secondary structure by the PIFtide hydrophobic motif. Neighboring regions, as well as the ATP binding pocket, catalytic loop, and activation loop, are also protected in the dimer, observations that support known conformational changes in AGC protein kinases upon hydrophobic motif pocket occupancy. Distinguishing between conformational changes induced by binding of the PIFtide motif into the hydrophobic pocket and those induced by dimerization is, however, not possible on the basis of this data.”

4. Because HDX rates are influenced by many factors beyond solvent accessibility, I recommend the substitution of “deprotection” for “exposure” in the HDX figure color code legends.

- We apologise for this confusion, and we believe the simplest way to represent this is to define this as increased or decreased exchange (as this has no issue of interpretation).

5. Typo? Figure 1. panel lettering (A-D) appears to omit panel A in the legend. The subsequent Fig. 1 panel letter designations in the legend would then need to be updated.

- We apologize for the missing figure legend, which was an oversight on our part after we realized that such a panel would be helpful to the reader. We have addressed this:

“Cartoon schematic of PDK1 domain architecture and construct boundaries of PDK1^{FL} (full-length) and PDK1^{SKD} (short kinase domain).”

6. Figure 4. This seems clear enough in the main text of the Results section, but it would be helpful to clarify- in the figure- which protein construct is the “reference” state (i.e., X% protection/deprotection with respect to which state?).

- We have added the annotation: “Reference: PDK1^{FL}” to Figure 4, since this is the reference state for both panels.

7. Regarding the XL-MS, I see that the dimer- and monomer-specific crosslinks were distinguished by resolving these states on SDS-PAGE. However, can this analysis confirm that the dimer-specific crosslinks are actually intermolecular? Couldn’t dimerization induce conformational rearrangements that favor new intramolecular crosslinks? My question here is mostly about the depiction in Fig. 2D and the interpretation “formed by exchange of the C-terminal PIFtide” (pg. 10).

- The reviewer makes an important point here, which we failed to clarify in the original manuscript. We note that the C-terminal PIFtide extension cannot physically reach the hydrophobic pocket of its own monomer. These cross-links therefore should be interpreted as being intermolecular, which is confirmed by their absence in the monomer reference and the resolution of a dimeric band by SDS-PAGE, which necessarily requires intermolecular cross-links. We have added the following sentence to the results to clarify this interpretation:

“We excluded the possibility of these cross-links being intramolecular on the basis that the linker sequence is too short for the PIFtide motif to bind into the hydrophobic pocket of its own protomer.”

Reviewer #2 (Remarks to the Author):

Activation of PDK1 by PIP3-driven autophosphorylation.
Levina et al, 2022.

This manuscript describes a model of PDK1 auto-activation that involves disengagement of the PH domain (by PIP3-binding) and homo-dimerisation that is driven by face-to-face contacts in the kinase domain and occupation of the PH-kinase linker in a groove on the adjacent protomer. Although there is no experimental structure of the PDK1-PDK1 interaction (affinity too low), the authors use alpha-fold and rosetta to predict the structure of the dimer and then use a number of different biochemical experiments (HDX, SEC-MALS, SAXS, activation assays, cross-linking) to support their model.

Given that PDK1 is a master kinase that activates a myriad of biological pathways, a better understanding of its regulation will have significant impact on our understanding of downstream kinase signalling cascades and this manuscript will be of great interest to those in the field. I recommend publication but just have some minor comments listed below.

- We appreciate the reviewer’s enthusiasm for our study and have addressed all the individual comments/criticisms below.

General comments:

There appears to be a lot more to the interface than Y288 and the model would be more convincing if further validated by other mutations in this area.

- We appreciate the reviewer’s desire for further mutational validation of the model, though we would respectfully argue that our characterization of the interface includes mutation of three independent motifs/surfaces of interaction (kinase domain dimerization (Y288A and Y288E), NYD motif (Y376A), and hydrophobic motif (F383A/M386A)), which are mutually compatible with our model.

While the biochemical data is strong, the manuscript would benefit from a cellular assay in which the different PDK1 constructs are tested in a more biological context. Are there well-established assays to monitor PDK1 activity in cells that could be used on their mutants?

- We can assure the reviewer that we tried very hard on this front, but have so far not been able to establish a robust assay for validating our model in cells. We have tried the following:

(a) we tried over-expressing PDK1 mutants in HEK293 cells with the goal of looking at FOXO3a localization as a readout. FOXO3a is a substrate of Akt, which itself is a substrate of PDK1. The principal problem with this assay is that the over-expression of PDK1 mutants with impaired activity compensates for the impaired activity. As such, the assay is entirely flawed.

(b) we obtained PDK1^{-/-} mESCs from Dr. Dario Alessi (MRC-PPU, Dundee) (Williams et al., 2000), in which we hoped to reconstitute wt PDK1 signaling (as was demonstrated) and compare our mutants. These cells, however, were not viable, neither in our hands or in theirs. As a gesture of good will, Dr. Alessi asked a colleague, Dr. Greg Findlay to remake the PDK1^{-/-} mESCs using modern CRISPR/Cas9 methods. Unfortunately, no viable clones were obtained, consistent with the known essentiality of PDK1. Presumably, the original PDK1^{-/-} mESCs had additional changes that allowed them to survive without PDK1, but these cannot now be reproduced.

(c) we teamed up with a group in-house to knock-in the mutations (Y288A, Y376A) F383A/M386A, and a kinase-dead control (D205A). Unfortunately, no viable clones could be obtained. Whilst this is consistent with them being critical to the activation of PDK1, in the absence of viable clones, we cannot verify that the editing occurred correctly. As such, this is not publishable and would need a conditional knock-in system. We believe that whilst extremely valuable, this is an entirely new study which needs considerable time to be executed properly.

In summary, whilst we share the reviewer's enthusiasm for a cellular readout, two years of failed attempts has taught us some valuable lessons. In retrospect, we were unlucky with the PDK1^{-/-} mESCs which were probably too old, but we also underestimated what it would take to do this for an essential gene. That will, naturally, be a focus going forward, but we believe it is important to share our findings with the scientific community as soon as possible, even without a cellular readout.

The authors use SAXS in figure 4. SAXS would be a powerful method to show a change in overall dimensions of the protein upon pip3 binding. Would it be feasible to add the pip3 headgroup (or other soluble pip3) and determine changes in radius of gyration or maximum interatomic distance to show disengagement of the PH domain from the kinase domain?

- In theory, the reviewer's suggestion is a very good one. However, we previously showed that the isolated headgroup of PIP₃ (IP₄) is unable to activate Akt1 (Lučić et al., PNAS 2018), suggesting that the membrane context is important for activation by lipid second messengers.
- To address the reviewer's question more directly, we now demonstrate that PIP₃ directly stimulates PDK1 kinase activity against a substrate peptide in vitro (Figure 5A-B), a result entirely consistent with all the biophysical and structural findings previously reported. We feel that this observation strengthens our model of PIP₃-mediated PDK1 activation much more than another biophysical method that would detect only the PIP₃-elicited conformational changes.

Much is made of the difference in K_d for FL vs PH binding to pip3 liposomes in Figure 4D, as well as differences in Hill-slope. However, a 2-fold difference in K_D is rarely significant and there are only a few points on these curves and certainly not enough to draw definitive conclusions of Hill-slope (especially for the PH domain, which looks like it would fit equally well to the standard hyperbolic saturation curve). A few more data points in the 0-5 μ M range would be very useful here.

- First of all, we must apologize for accidentally reporting an incorrect error for the K_d of binding of the PH domain. In fact, this value is 0.31 μ M, not 2.90 μ M, as was originally displayed on Figure 4D. We refer the reviewer to Supplementary Figure S8A, which shows the results of non-linear curve fitting with either (a) a one-site binding model or (b) specific binding with Hill slope. In each case, B_{max} was constrained to take a value < 1.0 as it cannot physically exceed 100% binding, and the fits compared. A statistical comparison of the two models in GraphPad led to the rejection of the null hypothesis (one-site binding model) in favor of binding with a Hill slope using the F-test. This is best visualized in the residuals of the fits to each model (Figure S8B). One should always be cautious about models, as they should also reflect a physical reality, not just a parameter that increases the goodness of fit. In this case, we believe there is evidence that a Hill coefficient of ~ 2 (we measured 1.91) could be explained by dimerization of the PH domain on membranes (Ziemba et al., *Biochemistry* 2013).
- Regarding the 2-fold difference in affinity, we agree that this is not large, but the question as to whether it is significant (statistically) or relevant (biologically) is a different one. A two-tailed t-test of the K_d derived from independent fitting of all three binding curves revealed values of $P=0.005$ for the difference in K_d and $P=0.030$ for the difference in n . We have added these values to the plot in Figure 4D, since we agree with the reviewer that the interpretation of this data is important to our final model for PDK1 activation by PIP3. It is also worth noting that our HDX-MS analysis shows sequestration of the PIP3 binding pocket in the intramolecular interface between the PH and kinase domains (Figure 4B-C). PIP3 binding therefore necessitates a conformational change to expose the binding site. The isolated PH domain, by contrast, has an accessible binding site. It therefore stands to reason that the affinity of the PH domain for PIP3 must be higher than FL PDK1. We have previously demonstrated that the same behavior is exhibited by both Akt1 (Ebner & Lučić et al, *Mol Cell* 2017, Truebestein et al., *PNAS* 2021) and Sgk3 (Pokorny et al., *JBC* 2021), for exactly the same reasons. We therefore maintain that this 2-fold difference in affinity is both significant and biologically meaningful.

Specific comments:

Figure 2

2B: colour scheme should be changed to be colour-blind friendly. More detailed main text

description of which construct (SKD-PIF/PIF-SKD) corresponds to which peak/curve in the figure for clarification.

- We have changed the color scheme of Figure 2B to make it color-blind friendly. Additionally, we have added in explicit references in the text to each peak, such that the reader immediately knows which peak we are referring to. We have also annotated Figure 2B with the shift in elution volume of SKD-PIF in the presence of ATP, since we discuss this in the text.

2E-H: Changes in HDX can be due to dimerization-induced protection from solvent or conformational change or PIF-induced protection from solvent. The authors are perhaps a bit too definite on which of these factors lead to protection of particular residues. Are they using their data to support their model, not their model to support their data? The observable is the rate of deuterium incorporation and the changes in that rate between the two constructs, it would be better to describe the HDX data as “supporting.....” rather than “reflecting.....”.

- We agree with the reviewer that our choice of language to describe our data and their interpretation was poor. The reviewer is entirely correct and we have modified the text extensively to present a more scientific analysis:

“In the N-lobe, the largest decrease in exchange is observed around the hydrophobic pocket (α C and α B helices), consistent with stabilization of the N-lobe secondary structure by the PIFtide hydrophobic motif. Neighboring regions, as well as the ATP binding pocket, catalytic loop, and activation loop, are also protected in the dimer, observations that support known conformational changes in AGC protein kinases upon hydrophobic motif pocket occupancy. Distinguishing between conformational changes induced by binding of the PIFtide motif into the hydrophobic pocket and those induced by dimerization is, however, not possible on the basis of this data.”

The α G helix should be labelled in the figure since it is an important component of the interaction and the fact that there is no change in HDX in this region is more than “surprising” as stated in the text. There does not need to be conformational change to protect deuterium incorporation when the region in question is in the middle of a dimer interface. More effort in the text to explain this unusual result is needed

- We agree with the reviewer and apologize for the missing labels in this figure. Whilst we were at first also ‘surprised’ by the absence of changes in the α G helix, we also realized that the main chain hydrogen bonds in the helix are satisfied entirely by helix formation. Prior structure determination of the kinase domain (with and without a Y288G mutation in the α G helix) indicates that the helix is rigidly associated with the rest of the domain. We were therefore satisfied that changes in the rate of deuterium incorporation should not necessarily be expected in this region. As such, the result is actually not as surprising as it may at first seem. Nevertheless, we have modified the text to better explain this result:

“Curiously, we did not observe any changes in the α G helix associated with

dimerization, which most likely reflects the fact that the α G helix is fully ordered in the monomeric kinase domain (Supplementary Figure 4C) as well as the kinase domain dimer model (Figure 1C) and, consequently, secondary structure changes in the α G helix upon dimerization are unlikely.”

Figure 4

4A: While I agree with the authors that comparing the phosphorylation kinetics of PDK1-FL and PDK1-SKD is misleading, perhaps a good control would be PDK1-FL F383A/M386A, where the HM has lost its ability to bind the hydrophobic pocket? This should behave like PDK1-SKD, based on figure 3E?

- The reviewer makes a good suggestion, which we also considered until we realized that it doesn't add any additional insight for the following reasons: (a) the LKD construct illustrates important residues for trans-autophosphorylation that are missing in the SKD construct (Figure 3A) (b) mutation of F383A/M386A abrogates this stimulatory effect (Figure 3E). We have therefore identified the machinery (which also includes Y376, Figure 3F) responsible for the higher catalytic machinery of PDK1-LKD and demonstrated that this effect arises from an interaction with the hydrophobic pocket (Figure 3D). Since it is questionable to what extent this additional mutant would add more insight, we focused our efforts more on demonstrating the direct activation of PDK1 by PIP3 (Figure 5A-B), which is a logical prediction and expectation from our findings.

Reviewers' Comments:

Reviewer #2:

Remarks to the Author:

The authors have made excellent alterations to the text that cover most of the concerns I raised in the initial review. However I still disagree with the authors regarding comments made on the Hill-slope, particularly for the PH domain alone. 5 points is simply not enough to define an accurate Hill-slope! In fact they estimate the Hill-slope as being 1.9 ± 1.4 (i.e 0.5 to 3.3) in their table yet state in the manuscript "The binding of the isolated PH domain could not be fit with a one-site binding model." This is simply incorrect given that the error estimate encompasses hill-slope=1 which it would be in a one-site binding model.

Showing a plot of residuals is cute but meaningless. If it is important for their model that there be positive cooperativity with a value of 2 then they need enough points on their curve to restrain the value to 2, not to between 0.5 and 3.3.

Just to clarify, I am not doubting that there is a difference in hill-slope between the PH domain and longer construct and am happy for that to be stated in the text. I am simply saying that there is not enough data to state that the Hill-slope of the shorter construct is 2.

Rebuttal to NCOMMS-21-44495-T

REVIEWERS' COMMENTS

Reviewer #2 (Remarks to the Author):

The authors have made excellent alterations to the text that cover most of the concerns I raised in the initial review. However I still disagree with the authors regarding comments made on the Hill-slope, particularly for the PH domain alone. 5 points is simply not enough to define an accurate Hill-slope! In fact they estimate the Hill-slope as being 1.9 ± 1.4 (i.e. 0.5 to 3.3) in their table yet state in the manuscript "The binding of the isolated PH domain could not be fit with a one-site binding model." This is simply incorrect given that the error estimate encompasses hill-slope=1 which it would be in a one-site binding model.

- We appreciate the reviewer's concerns here, which are entirely justified in the context of an error in the Hill slope of 1.4. Unfortunately, we have to apologize to the reviewer for reporting an incorrect error from the fit, which is in fact 0.44 and not 1.39 as we reported on the plot (I have no idea how this happened, because it was only reported on the figure and not in the text). Whilst, in principle, any data can be fit with any model, the goodness of fit is the important parameter. We believe that the data, while still only 6 points, strongly indicates positive cooperativity. The precise value of the Hill coefficient is subject to a similar error as that obtained for FL PDK1 (0.44 vs 0.42 respectively). This indicates that the Hill coefficient could take a value anywhere between 1.47 and 2.35, which excludes a single site binding model with a Hill coefficient of 1 (which is also evident from the Supplementary Figure 8A we reported in which a one-site binding model was used to fit the PH domain binding). As we reported, this finding is consistent with a previous report of PH domain dimerization on membranes containing PIP3 (Ziemba et al., 2013), but we do acknowledge that without further data points, it is not possible to constrain the Hill coefficient to a precise value of 2. Since the error values are important for interpretation of the data, we have added them into the main text so as to aid the reader.

"The binding of the isolated PH domain could not be fit with a one-site binding model (Supplementary Fig. S8A-B), but exhibited positive cooperativity (Hill coefficient = 1.91 ± 0.44), consistent with dimerization of the isolated PH domain on the membrane, as has previously been reported¹⁴. However, PDK1^{FL} exhibited even stronger positive cooperativity, with a Hill coefficient of $6.01 (\pm 0.42)$, consistent with additional interactions that stabilize the PIP₃-bound conformation, including kinase domain dimerization, NYD motif interactions, and HM exchange."

Showing a plot of residuals is cute but meaningless. If it is important for their model that there be positive cooperativity with a value of 2 then they need enough points on their curve to restrain the value to 2, not to between 0.5 and 3.3.

Just to clarify, I am not doubting that there is a difference in hill-slope between the PH

domain and longer construct and am happy for that to be stated in the text. I am simply saying that there is not enough data to state that the Hill-slope of the shorter construct is 2.

- The precise value of the Hill coefficient for the PH domain is not important to our model. What is more important is the difference in Hill-slope between the PH domain and FL PDK1, because this confirms the additional interactions that occur upon kinase domain dimerization in the context of the full-length protein. We appreciate that the reviewer agrees with our conclusions regarding this point.